# Cellular events of acute, resolving or progressive COVID-19 in SARS-CoV-2 infected non-human primates

M. D. Fahlberg [1], R. V. Blair[1,2], L. A. Doyle-Meyers[1,3], C. C. Midkiff[1], G. Zenere[1], K. E. Russell-Lodrigue [1,3], C. J. Monjure[1], E. H. Haupt[1], T. P. Penney[1], G. Lehmicke[1], B. M. Threeton[1], N. Golden[1], P. K. Datta[1,2], C. J. Roy [1,4], R. P. Bohm[1,3], N. J. Maness [1,4], T. Fischer [1,4], J. Rappaport[1,4] & M. Vaccari [1,4✉]

Understanding SARS-CoV-2 associated immune pathology is crucial to develop pan-effective vaccines and treatments. Here we investigate the immune events from the acute state up to four weeks post SARS-CoV-2 infection, in non-human primates (NHP) with heterogeneous pulmonary pathology. We show a robust migration of CD16 expressing monocytes to the lungs occurring during the acute phase, and we describe two subsets of interstitial macrophages (HLA-DR$^+$CD206$^-$): a transitional CD11c$^+$CD16$^+$ cell population directly associated with IL-6 levels in plasma, and a long-lasting CD11b$^+$CD16$^+$ cell population. Trafficking of monocytes is mediated by TARC (CCL17) and associates with viral load measured in bronchial brushes. We also describe associations between disease outcomes and high levels of cell infiltration in lungs including CD11b$^+$CD16$^{hi}$ macrophages and CD11b$^+$ neutrophils. Accumulation of macrophages is long-lasting and detectable even in animals with mild or no signs of disease. Interestingly, animals with anti-inflammatory responses including high IL-10: IL-6 and kynurenine to tryptophan ratios show less severe illness. Our results unravel cellular mechanisms of COVID-19 and suggest that NHP may be appropriate models to test immune therapies.

[1] Tulane National Primate Research Center, Covington, LA, USA. [2] Department of Pathology and Laboratory Animal Medicine, Tulane University School of Medicine, New Orleans, LA, USA. [3] Department of Medicine, Tulane University School of Medicine, New Orleans, LA, USA. [4] Department of Microbiology and Immunology, Tulane University School of Medicine, New Orleans, LA, USA. ✉email: mvaccari@tulane.edu

The COVID-19 pandemic is a major public health crisis, causing a global medical emergency.

There are currently no effective vaccines or drugs approved to completely treat or prevent COVID-19, and much remains unknown about the pathogenesis of the disease. COVID-19 presents with highly variable outcomes, and patients that develop symptoms exhibit a wide range of disease, spanning from mild (fever, cough, shortness of breath) to severe (dyspnea, pneumonia, rapidly progressing radiographic changes)[1]. It is currently estimated that up to 15% of COVID-19 patients progress to acute respiratory distress syndrome (ARDS)[2], which is the major cause of death among fatal SARS-CoV-2 cases. Studies elucidating molecular and immunological details of SARS-CoV-2 infection are underway, and many of the reported clinical observations point to the disease being at least partly the result of an excessive host response aimed to clear the virus but instead contributing to disease development[3]. This hypothesis is supported by studies on the closely related coronavirus SARS-CoV-1[4–6], and by clinical reports showing elevated levels of the pro-inflammatory cytokine IL-6 in SARS-CoV-2 infected patients, and profound dysregulation of the myeloid cell compartment, particularly among those with severe symptoms[3,7–9].

Interestingly, the increase in IL-6 coincides with the upregulation of chemokines responsible for myelopoiesis and monocyte recruitment to the lungs[3,10] suggesting that peripheral inflammatory monocytes and tissue macrophages may have a role in the cytokine storm seen in severe COVID-19 patients[11]. The underlying inflammatory state and increased myelopoiesis seen in older individuals may be responsible for the high mortality rate seen in this group[12]. A better understanding of the immunopathology of COVID-19 will aid the development of targeted immunotherapies and effective vaccines.

Non-human primates (NHP) have been used as animal models to study acute respiratory diseases such as SARS and MERS[13]. In these studies, African green monkeys (AGM) have been shown to be more susceptible to severe disease than rhesus macaques (RM)[14]. Regardless of species differences, and given the overlap between NHP and with human immune systems, both NHP species have been useful models to study host responses to coronaviruses[5]. We and others have shown that both AGM and RM can be successfully infected with SARS-CoV-2 via the intranasal and the intratracheal route, or by aerosol[15–18]. Following exposure, AGM and RM have high levels of virus replication in the respiratory tract for up to 2 weeks, and both species develop pulmonary pathology characteristic of COVID-19, that varies from very mild to severe[19]. Other groups have described the very early immune events in SARS-CoV-2 infected rhesus macaques, with perturbation of monocytes populations, increased macrophages in lungs, and cellular activation in blood all happening a few days following the infection[16].

Here, we extend on these findings by describing the kinetics of immune events following the infection, spanning from the acute phase up to 4 weeks following infection, in the blood and the lungs of four African green monkeys (AGM) and four rhesus macaques (RM) with a heterogeneous spectrum of COVID-19. Overall, we observed a multiphasic mobilization of innate and adaptive cells from the blood to the lung that is strikingly common to both species. The acute inflammatory phase was characterized by an increase in the frequency of monocytes that was positively associated with bronchial viral replication. CD11b+ macrophages and neutrophils were detected in the lungs of all infected animals, however, with greater numbers in severe disease. Consistent with the resolution phase of inflammation in the lung[20], the second phase was characterized by a switch toward type 2 responses in both species. We identified two populations of interstitial macrophages (HLA-DR+CD206−) in BAL; (1) CD11c+CD16hi population increasing during the acute phase of infection that was associated with IL-6 levels in the plasma, and (2) a long-lasting CD11b+ and CD16hi population accumulating in lungs of both AGM and RM, particularly in those with increased disease severity. Animals with more pronounced clinical signs had a higher level of infiltrates in lungs when compared with animals with no or mild signs, lower IL-10:IL-6 and kynurenine /tryptophan ratios (anti-inflammatory: pro-inflammatory) cytokines in plasma[21]. By investigating temporal and spatial changes in immune cells our study links together findings in both NHP and human patients and unravels cellular events of COVID-19[22,23]. Further, we show that NHP are highly relevant models to study COVID-19 immunopathology and suggest that they may be appropriate models to test immune therapies and vaccines for SARS-CoV-2.

## Results

**Increased levels of monocytes and chemokines after the infection is associated with viral load and disease severity.** Four adult RM and four adults AGM were exposed to SARS-CoV-2 by either aerosol or mucosal challenge including buccal, intranasal, intratracheal, and conjunctival (multi-route) exposure (Supplementary Table 1). The study concluded at 4 weeks post-exposure (Fig. 1a)[18]. All eight animals had detectable viral load by RT-PCR on nasal and pharyngeal swabs and bronchial brushes by week 1 (Fig. 1b and Supplementary Table 2). Viral RNA was also detected in pharyngeal, nasal, buccal, rectal and vaginal swabs. No differences in viral load (VL) were observed between the different routes of exposure or between the two species at week 1 (Supplementary Table 2 and Fig. 1b). Of note, by week 2 AGM had higher VL than RM in the bronchial brushes and by week 3 VL in RM was mostly undetectable while it remained above detection levels in AGM until study completion (Fig. 1b and Supplementary Table 2). Similar to humans, the course of disease varied widely across animals, as detailed in Table 1. Of the four AGM enrolled, two animals developed severe respiratory signs and were euthanized on days 8 (NC34) and 22 (NC33) post-infection, following established endpoint criteria for NHP study protocols, and as we previously reported[18]. Of the two remaining AGM, NC38 had multifocal mild to moderate interstitial pneumonia scattered throughout all lung lobes and NC40 was mostly asymptomatic and had scant inflammation in all lung lobes at necropsy. Three of the four RM (GH99, HD09, and FR04) developed pneumonia characterized by a granulomatous to pyogranulomatous inflammatory response. This was severe in GH99, mild in HD09 and minimal in FR04. Histopathology of the fourth RM (HB37) revealed a lymphocytic vasculitis and proliferative vasculopathy in the right middle lung lobe.

Clinical signs observed during the course of the study included mild tachypnea (day 11) and cough (day 15) in GH99 (Supplementary Table 3). Intermittent mild to moderate pyrexia was observed in HB37 (days 1–21 post-infection). Viral loads in the bronchial alveolar brushes at all timepoints, temperature, pulse oxygen levels (SpO2) and ranking did not differ between female ($n = 3$) and males ($n = 5$) or between animals exposed to different routes of infection (multi-route $n = 4$ and aerosol $n = 4$), possibly due to the small number of animals per group (Supplementary Table 2–4).

Viral load was not significantly associated with pulmonary pathology, however, we observed an inverse association between viral load at week 1 and disease severity ranking (1–8 from most severe to least severe) based on the above observations and histopathology findings (AGM: $R = 0.8$; RM: $R = -0.4$, all animals: $P = 0.057$, $R = -0.71$, by the Spearman correlation test) (Fig. 1c and Supplementary Fig. 1a, b, Supplementary Table 5).

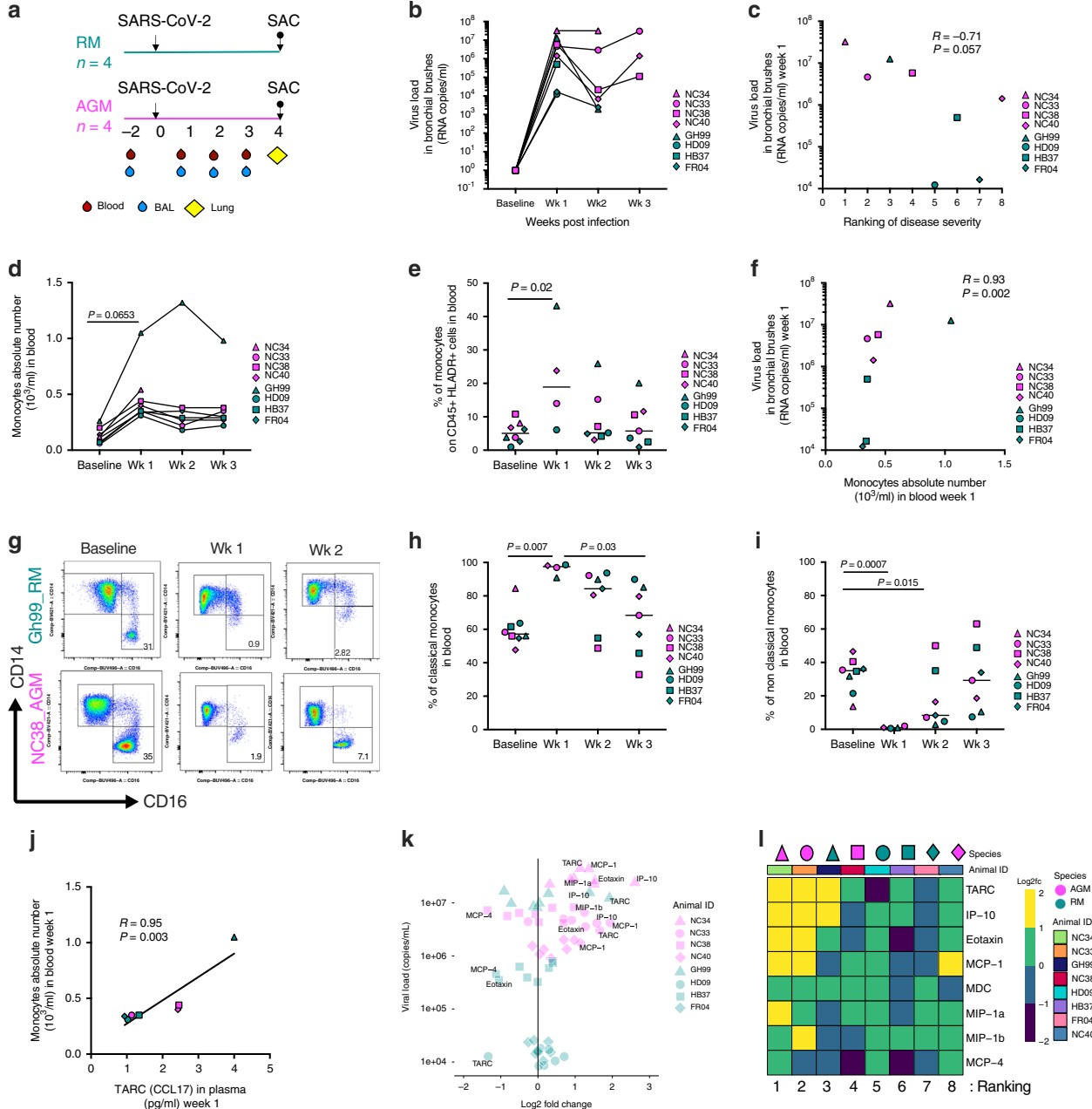

**Fig. 1 Monocytes are a correlate of bronchoalveolar viral replication. a** Schematic of study design. Arrows represent time of inoculation with SARS-CoV-2 (day 0) or sacrifice (up to 28 days post-infection). Unless stated otherwise, rhesus macaque data are shown in teal, and African green monkey data are shown in magenta. **b** SARS-CoV-2 RNA levels in bronchial brushes over time. **c** Association between SARS-CoV-2 load in bronchial brushes and ranking of disease severity in all eight animals (most severe case = 1; mildest case = 8) (Spearman test, $n = 8$). **d** Changes in monocyte count in blood before and after infection (by one-way ANOVA adjusted, for multiple comparisons by Tukey's test, $n = 8$). **e** Proportion of total monocytes (HLA-DR$^+$ CD16$^+$ and/or CD14$^+$ out of CD45$^+$ leukocytes) in PBMCs over time (baseline and week 1 = 8 animals, week 2, 3 = 7 animals, Kruskal–Wallis test, Dunn's multiple comparisons). **f** Correlation between SARS-CoV-2 viral load in bronchial brushes and absolute count of monocytes in blood at 1-week post-infection (1wk pi) ($n = 8$). **g** Representative flow cytometry changes in non-classical monocytes (CD14$^{low}$ CD16$^{hi}$) after the infection in the blood (1wk pi). **h** Percent of classical (CD14$^{hi}$ CD16$^{low}$) and **i** non-classical monocytes (CD14$^{low}$ CD16$^{hi}$) out of total monocytes in PBMCs (baseline = 8 animals, week 1 = 4 animals, week 2, 3 = 7 animals; Kruskal–Wallis test, Dunn's multiple comparisons). **j** Correlation between monocytes absolute numbers in blood and TARC (CCL17) in plasma (1wk pi). **k** Relationship between fold-change of chemokines and SARS-CoV-2 viral load in bronchoalveolar brushes (1wk pi). **l** Heatmap displaying log2 fold-change of eight chemokines involved with traffic of monocytes to the lung (1wk pi). Animals are ordered by ranking of disease severity. All correlations analyses are by the Spearman correlation test.

Interestingly, virus-driven immunological changes were remarkably similar between species at this time (Fig. 1 and Supplementary Fig. 1). To investigate overall changes driven by viral replication in both groups, we combined their immunological analysis, however, at the same time, we maintained different specifications for each species (teal = RM and magenta = AGM). Data and statistics are also shown separately per group in Supplementary Fig. 1. At 1 week after infection, both AGM and RM had an increase in the absolute number and frequency of monocytes in the blood ($P = 0.065$, and $P = 0.02$, respectively, by

**Table 1 Heterogeneous clinical and pathological outcomes following SARS-CoV-2 infection.**

| Animal | Clinical findings | Clinical score | Pathology findings | Pathology score | Ranking |
|---|---|---|---|---|---|
| NC34 | Severe reduction in $SpO_2$, moderate tachypnea, and severe hypothermia just prior to euthanasia for severe respiratory distress at 8 days post-infection. Less than 24 h earlier, the only abnormality was a moderate reduction in $SpO_2$. Radiography appeared unremarkable until euthanasia when moderate to severe alveolar pattern with a lobar sign and air bronchograms in right lung fields was observed | 9 | Bronchointerstitial lung pattern in all lobes that was most severe in the right caudal lung lobe. Microscopically all lung lobes exhibited similar changes with the right caudal and right accessory lung lobe being the most severely affected. The lung lesions were comprised of diffuse alveolar damage with hyaline membrane formation, type II pneumocyte hyperplasia, and rare multinucleated giant syncytia. Some regions showed early evidence of organizing pneumonia and fibrosis | 19 | 1 |
| NC33 | Severe reduction in $SpO_2$, severe tachypnea, and severe hypothermia just prior to euthanasia for severe respiratory distress at 22 days post-infection. Less than 24 h earlier, the only abnormality was a mild reduction in $SpO_2$. Radiography appeared unremarkable until euthanasia when moderate to severe alveolar pattern in right middle/caudal lung fields and mild alveolar pattern left caudal lung fields were observed | 10 | Severe diffuse alveolar damage of the right lower lung lobe. Other lobes exhibited suppurative alveolar infiltrate with lesser fibrin and edema. The right lung lobe contains peracute to acute lesions of necrosis, fibrin, and hemorrhage with minimal cellular infiltrate | 14 | 2 |
| GH99 | Transient cough beginning on 15 days post-infection. Radiography showed significant changes at 11 days post-infection which improved through 27 days post-infection | 1 | Severe bronchopneumonia of the right lower lung with granulomatous to pyogranulomatous inflammatory response | 10 | 3 |
| NC38 | Minor intermittent reductions in $SpO_2$ | 0 | Mild interstitial pneumonia with low numbers of multinucleated giant cells and rare atypical pneumocyte hyperplasia | 8 | 4 |
| HD09 | Mild focal increase opacity ventral lung field at 11 days post-infection | 1 | Minimal to mild inflammation scattered throughout the lungs | 9 | 5 |
| HB37 | Intermittent mild to moderate pyrexia from 1 to 21 days post-infection with mild dehydration from 14 to 21 days post-infection | 1 | Proliferative lymphocytic vasculitis and proliferative vasculopathy in the right middle lung lobe. Scattered interstitial inflammation throughout the remaining lung lobes | 5 | 6 |
| FR04 | Unremarkable | 0 | Bronchioles exhibited multifocal, mild mineralization | 6 | 7 |
| NC40 | Minor intermittent reductions in $SpO_2$ | 1 | Minimal changes within the lungs. Mild multifocal thickening of the alveolar septa in the right lower lobe associated with a mild inflammatory infiltrate | 4 | 8 |

Clinical findings are summarized for the duration of infection. Clinical scores, pathology findings, and pathology scores reflect measurements at the time of necropsy. Rankings were determined based on a combination of the clinical findings and scores throughout the duration of infection and the pathology findings and scores at the time of necropsy (1 = severe; 8 = mild).
$SpO_2$ = oxygen saturation.

the Tukey multiple comparison tests, $n = 8$) compared to baseline levels (Fig. 1d, e). This change was significant in AGM alone (absolute number: $P = 0.0006$, $n = 4$), and was also observed in all RM, particularly GH99 (Supplementary Fig. 1c–f). Of note, the increase in absolute number of monocytes remain significant when GH99 was removed from the statistical analysis ($P = 0.00002$, Supplementary Fig. 1g). The absolute number of neutrophils in blood did not change over time (Supplementary Fig. 1h). Interestingly, the absolute number of monocytes at week 1 was strongly positively associated with the viral load levels in the bronchial brush samples ($P = 0.002$, $R = 0.93$ by the Spearman test) when all the animals were combined (Fig. 1f) (AGM: $R = 0.8$; $R = 1$, not significant in separated groups, Supplementary Fig. 1i, j), and remained significant when GH99 was removed from the statistical analysis (Supplementary Fig. 1k).

We then used the CD14 and CD16 markers to discriminate between three circulating monocytes subtypes: classical (CD14$^{hi}$ CD16$^{low}$), intermediate (CD14$^{hi}$ CD16$^{hi}$) and non-classical (CD14$^{low}$ CD16$^{hi}$) within the total (CD45, live, HLA-DR$^{+}$ CD16hi) monocyte population by flow cytometry (Fig. 1h–i and Supplementary Fig. 2). Subsets were gated as shown in Supplementary Fig. 2a, using a strategy that excludes CD16$^{+}$ NK cells that are negative for HLA-DR[24]. At week 1, the frequency of classical monocytes was significantly increased in the blood, as seen in humans[25], and the non-classical population was profoundly depleted (increased levels baseline versus week 1: classical $P = 0.007$ and decreased non-classical $P = 0.005$ in all animals combined, $n = 4$) (Fig. 1h, i and Supplementary Fig. 2b–e). As we observed for the frequency, the absolute counts of classical and non-classical monocytes at week 1 were also significantly increased and decreased, respectively, compared to the baseline values (Supplementary Fig. 2f, g). Interestingly, pre-infection levels were never completely restored even after 3 weeks post-infection (week 1 vs. week 3: classical $P = 0.03$ and non-classical $P = 0.03$), as the number of circulating CD16$^{hi}$ monocytes increased over time (non-classical week 1 versus week 3: $P = 0.0019$). These results suggest emerging myelopoiesis and rapid and robust recruitment of patrolling monocytes in tissues following infection in both species. We did not observe changes in the frequency or number of intermediate monocytes (Supplementary Fig. 2h, i). In line with what we observed for the total monocyte count, at week 1 the counts for the subsets of monocytes were positively associated with viral replication detected at the bronchial alveolar sites (classical: $R = 0.98$; $P = 0.06$, non-classical: $R = 0.96$; $P = 0.03$ and intermediate $R = 0.94$; $P = 0.06$, by the Pearson test), further suggesting that the robust myelopoiesis is therefore driven by the virus in the lung. Within the total monocytes, the frequency of non-classical monocytes was significantly associated with lower viral loads (classical: $R = -0.9867$; $P = 0.0133$, by the Pearson Test, adjusted for multiple comparisons $P = 0.07$). It is important to note that none of these analyses were significant when the values were adjusted for multiple comparisons (Supplementary Table 6).

Recent studies in humans have described an increase in the frequency of CD14 positive cells expressing low levels of HLA-DR in COVID-19 patients. These cells phenotypically resemble suppressive subsets of myeloid-derived suppressor cells (MDSCs)[8] thus they may decrease disease-driven inflammation. We and others have previously identified MDSCs in the blood of monkeys by flow cytometry as HLA-DR$^{low}$ CD14$^{hi}$ cells within the CD45/live population[26,27]. Some of the animals in our study showed increased MDSCs frequency at week 3 compared to baseline levels, however, these changes were not significant (Supplementary Fig. 2l). We measured Arginase 1 activity as a functional assay to assess the function of immunosuppressive myeloid cells at the same time points (baseline and week 3) and

we did not observe significant changes in plasma over time (Supplementary Fig. 2m)[28].

We analyzed a group of eight chemokines known to be involved in the mobilization of innate cells, including monocytes, from the blood to the lung: IP-10 (CXCL-10), MCP-1 (CCL2), MCP-4 (CCL13), Eotaxin, TARC (CCL17), MDC (CCL22), MIP-1α, and MIP-1β (Supplementary Fig. 3a–h). Some of the chemokines measured at baseline in plasma differed between the two species (MCP-4, Eotaxin, MIP-1b, and MDC) as shown by the principal component analysis in Supplementary Fig. 3i (Supplementary Tables 7–9). Plasma levels of TARC (CCL17), a ligand for CCR4 that is involved in Th2 cell recruitment[29], were strongly associated with the frequency of monocytes in the blood ($P = 0.003$, $R = 0.95$, Spearman test, adjusted for multiple comparisons: $P = 0.02$) (Fig. 1j). Similarly, TARC/CCL17 was also associated with viral replication in BAL at week 1 (Fig. 1k and Supplementary Fig. 3j, k) ($P = 0.02$, $R = 0.857$, adjusted for multiple comparisons: $P = 0.2$, and $R = 0.927$ and $P = 0.016$ when GH99 is excluded from the analysis), as was IP-10 ($P = 0.05$, $R = 0.78$, adjusted for multiple comparisons) when all animals were grouped together (Fig. 1k and Supplementary Fig. 3l), a ligand for CXCR3 that is associated with disease severity and predicts progression of COVID-19 in humans[30]. To gain a better understanding of a common milieu possibly driving monocyte migration observed in both groups we analyzed changes in the levels of these markers with respect to the baseline (Fig. 1l and Supplementary Table 8). Overall this analysis revealed that animals with the most severe disease outcomes had higher levels of chemokines at week 1, including IP-10, TARC (CCL17).

**Myelomonocytic cells infiltrate BAL and are associated with IL-6**. Influx of macrophages has been observed in the lung of severe cases of COVID-19 in humans[31]. To monitor cell trafficking into the lungs, we collected bronchoalveolar lavage (BAL) at baseline, and after each week following infection, up to week 3. As expected, the majority of the cells found in BAL at baseline were alveolar macrophages (AM)[32]. These cells are readily identified by the mannose receptor C type 1 (CD206) marker, express high levels of HLA-DR, and are positive for the scavenger receptor CD163 (Fig. 2 and Supplementary Fig. 4a, b). We observed a decrease in the proportion of AM at week 1 post-infection in animals of both species, corresponding with an influx of myeloid HLA-DR$^{+}$CD206$^{-}$CD163$^{-}$ cells (Fig. 2a, b and Supplementary Fig. 4c). Levels of AM similar to pre-infection were restored at week 2 in all but one animal (GH99), and a concomitant increase of CD86 mean fluorescent intensity (MFI) in this population was observed suggesting an M1 activation state[33] (Supplementary Fig. 4d). An increase was also seen in the frequency of CD206$^{+}$ CD163$^{-}$ that have been previously described as endothelial cells, which facilitate lymphocyte trafficking and they are detectable in BAL during inflammatory events (Fig. 2c)[34].

T-distributed stochastic neighbor embedding (tSNE) analysis revealed changes in the phenotype of CD16, CD11b, and CD11c positive populations most prominently at 1–2 weeks post-infection (Fig. 2d–f). In contrast to what we observed in the blood, there was an increase of CD16$^{+}$HLA-DR$^{+}$ cells in BAL overtime (Fig. 2g, and Supplementary Fig. 4e, f). We observed a transient increase in the level of CD11c$^{+}$CD16$^{+}$ myelomonocytic cells, which function has been described as patrolling, and they participate in rapid tissue invasion and monocyte survival (baseline vs. week 1: $P = 0.015$) (Fig. 2h)[35,36]. The frequency of CD11c$^{+}$ macrophages in BAL at week 3 was strongly associated with the levels of the pro-inflammatory IL-6 cytokine in the plasma at the same time point (week 3: $P = 0.003$, $R = 0.96$, Spearman test) (Supplementary Fig. 5). The CD11b marker is

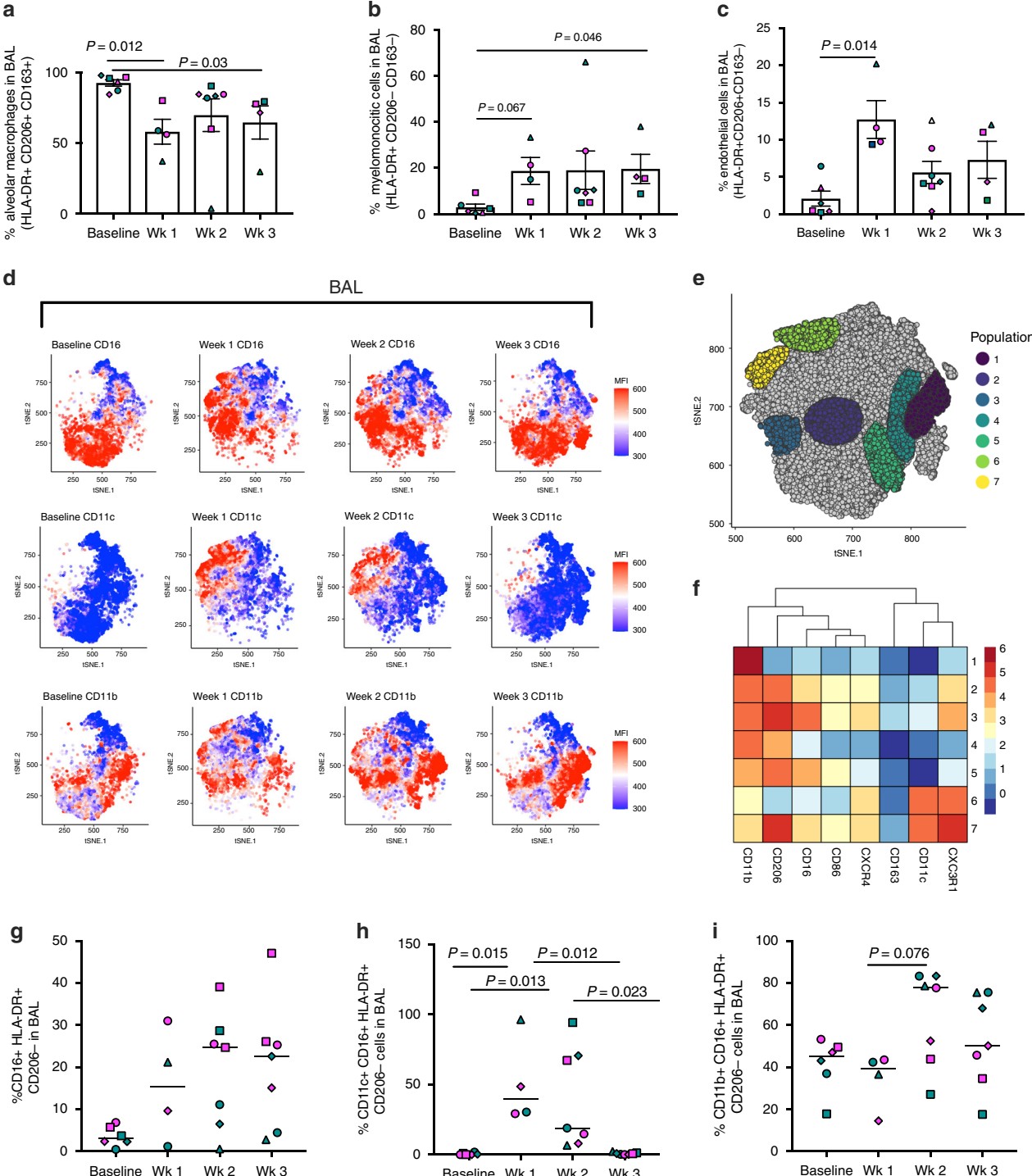

**Fig. 2 Myelomonocytic cells increase in BAL is associated with IL-6. a** Decrease of alveolar macrophages (defined as HLA-DR+ CD206+ CD163+ cells) in the BAL after infection. **b** Increase in the frequency of myelomonocytic infiltrates (defined as HLA-DR+ CD206− CD163−) and **c** of endothelial cells (defined as HLA-DR+ CD206− CD163+) at infection in BAL (1wk pi). For figure **a**–**c**, n = 7, 4, 8, 4 for baseline weeks 1, 2, and 3 time points, respectively, and analyzed by the Kruskal–Wallis test, Dunn's multiple comparison test. The bars represent mean and standard error. **d** tSNE plots displaying kinetics of expression of CD16, CD11b, and CD11c markers on live/ HLA-DR+ cells populations in BAL before and after infection. **e** Gating of CD11c+ and CD11b+ populations on the tSNE map based on antigen expression and discrete density clustering. **f** Characterization of populations shown in **e**. The heatmap depicts the fold-change values of each antigen compared to the channel values of negative-expressing cell populations. **g** Increase in the percentage of CD16+ HLA-DR+ cells and **h** in the percentage of interstitial macrophages (CD16+ CD206− HLA-DR+) expressing of CD11c and **i** CD11b in BAL overtime (baseline n = 8, week 1 n = 4, week 2 and 3 n = 7; Kruskal–Wallis test, Dunn's multiple comparisons).

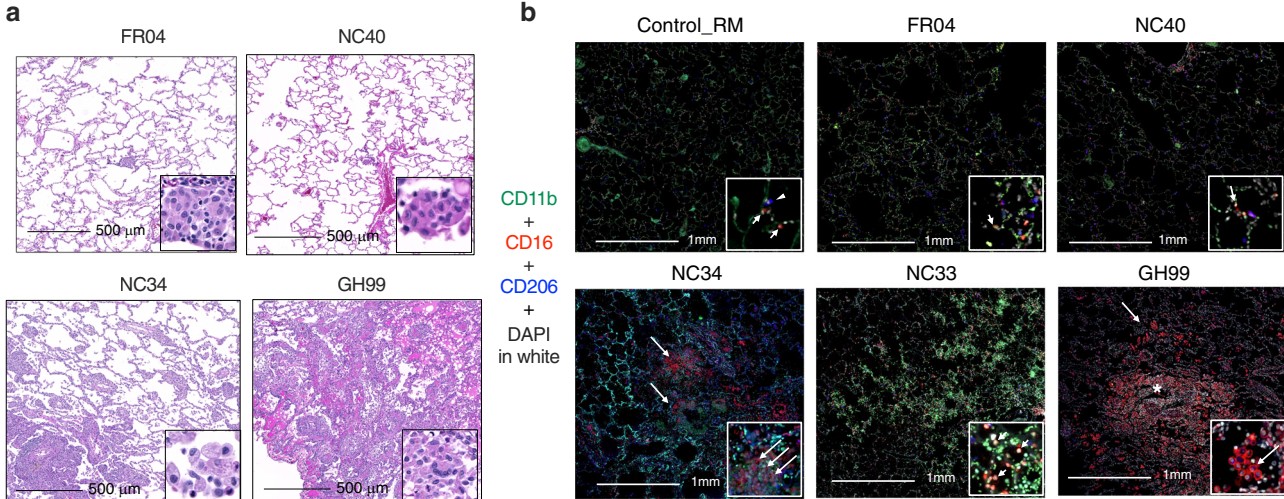

**Fig. 3 Interstitial macrophages are increased in the lung of SARS-CoV-2 infected animals. a** Macrophage infiltration varied from minimal to severe in both rhesus and AGM infected with SARS-CoV-2. **b** Fluorescent immunohistochemistry in the lung of AGM and RM showing infiltration of CD16+ CD11b+ CD206− macrophages (arrows), with lesser numbers of CD16+ CD206+ macrophages (arrowheads). Inset showing a higher magnification of CD16+ CD11b+ CD206− macrophages (arrows). Asterix shows macrophages multifocally surrounding small airways in GH99. White: DAPI (nuclei); green: CD11b; red: CD16; blue: CD206.

required for monocyte migration to sites of inflammation[37], and is highly expressed on myeloid cells recruited from the blood and differentiating into macrophages[38]. We also observed an increase in the percentage of CD11b+CD16+HLA-DR+CD206− myelomonocytic cells in BAL at weeks 2 and 3 following infection (Fig. 2i), a population that is involved in regulating lung inflammatory response[20,33,39,40]. The gating strategy for the two myelomonocytic cells populations is shown in Supplementary Fig. 4g.

**Interstitial macrophages accumulate in the lungs of infected animals.** Lungs were collected for all animals at the end of the study, corresponding to 4 weeks post-infection (1 week following the last BAL collection shown in Fig. 2), with the exception of NC33 and NC34. These two animals had higher levels of IL-6 at necropsy compared to the other six monkeys, and NC34 had symptoms similar to ARDS[18] (Fig. 3). Overall, the infiltration of macrophages was observed in the lungs of SARS-CoV-2 infected AGM and RM at necropsy and was more robust in animals with severe disease (Fig. 3a). We further characterized the macrophages population in the lung by staining with the AM marker CD206 (in blue) and with CD11b integrin (in green) and CD16 (red) (Fig. 3b). Low numbers of macrophages were seen in the lungs of a non-infected control RM, scattered throughout the interstitium and alveolar spaces, as shown by the arrows. In the control, this population was bi-morphic and composed of CD11b negative CD16+CD206− (arrow) in the alveolar septa or CD16+ CD206+ cells within the alveolar spaces (arrowhead) (Fig. 3b). In all animals that were infected, we could detect CD11b+CD16+ macrophages. FR04 (RM ranking = 7) lungs had macrophages scattered throughout the interstitium and alveolar spaces. The phenotypic population was heterogeneous and contained low numbers of CD11b+ CD16+ CD206− macrophages. The most severe RM, GH99 (ranking = 3), had large numbers of macrophages multifocally surrounding small airways denoted by the asterisks. Of note, this animal had a higher absolute number of monocytes in the blood at week 1 (Supplementary Fig. 1d). Low numbers of CD11b+ CD16+ CD206− cells were scattered throughout the predominant population of CD11b+ CD16+ CD206+ macrophages. Macrophages were also

scattered throughout the interstitium and alveolar spaces. NC34 and NC33 (AGM ranking = 1 and 2, respectively) had multifocal to coalescing areas of hemorrhage (NC33 indicated by the arrows), and pulmonary parenchyma infiltrated by multifocal aggregates of macrophages (NC34) with infiltration by low to moderate numbers of macrophages. The phenotypic population was heterogeneous and composed of moderate numbers of CD11b+ CD16+ CD206− macrophages.

Aggregates were determined by staining animals FR04, GH99, and NC34 with CD68/CD163 (red) and CD3: (blue). Animal FR04 (RM ranking = 7) had isolated CD3+ cells scattered throughout the lung. GH99 (RM ranking = 3) had cell aggregate with mixed populations of CD3+ and CD68/CD163+ cells in an isolated area of inflammation. NC34 (AGM ranking = 1) had rare, small aggregates of CD3+ cells (Supplementary Fig. 6). Altogether these findings present a picture of long-lasting lung inflammation that, albeit more pronounced in animals with severe signs of disease, is also present in animals with no detectable viral load and mild or no signs of disease after 4 weeks from the start of an infection.

**Neutrophils accumulate in animals with severe disease.** In this study, we did not observe changes in the absolute number of neutrophils in blood during the course of the infection (Supplementary Fig. 1h). Interestingly, studies in monkeys and humans showed increased levels of neutrophils and suggested deregulation of their maturation, with CD11b being one of the markers defining pre-neutrophils[8,9,41–43]. We stained lungs from animals NC33, NC34, GH99, FR04, and NC40 with anti-myeloperoxidase (MPO) antibody for neutrophils and with anti-CD11b antibody, and we included two controls, one RM and one AGM (Fig. 4a). MPO+ cells[44] were elevated in animals with severe disease (NC33, NC34). GH99 had areas with neutrophil accumulation and the remaining animals were low. Controls had very few neutrophils in the lungs (Fig. 4b). CD11b+ MPO+ neutrophils were only present in infected animals and were also high in NC33 (Fig. 4c). Accordingly, neutrophils and lymphocytes could be seen in animals with severe disease but not in animals with moderate disease (Supplementary Fig. 7). NC34 (necropsy day 8 post-infection; rank = 1, severe) had lymphocytic aggregates that were

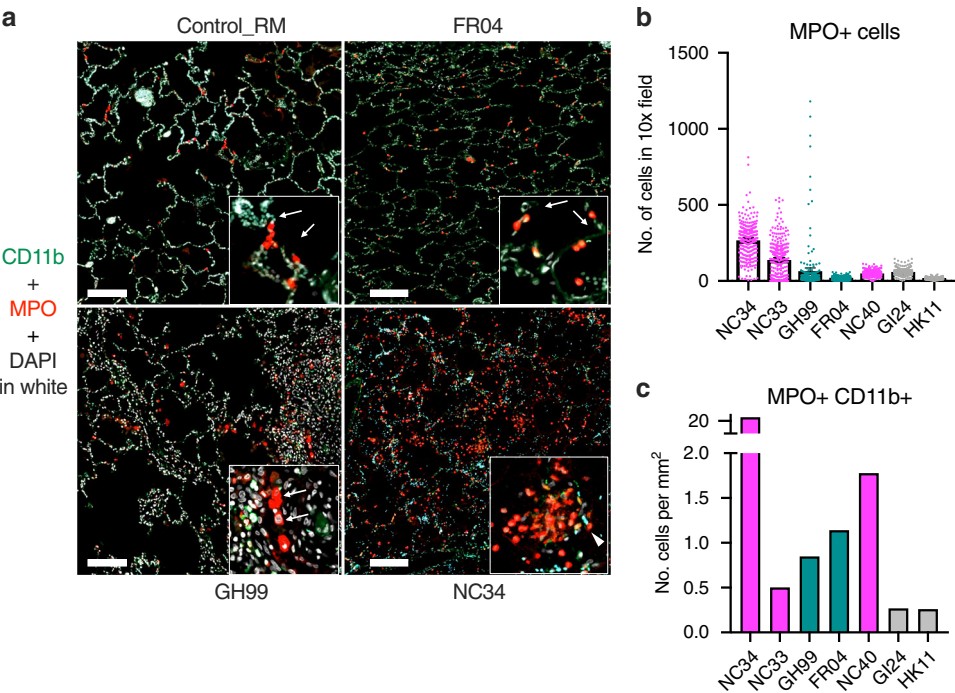

**Fig. 4 CD11b$^+$ neutrophils accumulate in the lung of infected animals with severe disease. a** Fluorescent immunohistochemistry in the lung of AGM and RM showing infiltration of MPO$^+$ neutrophils and MPO$^+$ CD11b$^+$ neutrophils. MPO and CD11b staining. White: DAPI (nuclei); green: CD11b; red: MPO. The number of MPO positive cells (red, arrows) increases in proportion to the severity of pulmonary disease from rare in the control (HK11), to mild in FR04, moderate in GH99 and severe in NC34. Dual positive MPO$^+$ CD11b$^+$ positive cells (arrowheads) are present in the most severely affected animals (NC34). **b** Mean and number of MPO$^+$ cells, each dot is one count the standard deviation represents variability in the number of positive cells/area and **c** and MPO$^+$ CD11b$^+$ per mm$^2$ in lung necropsy.

perivascular and intermixed with neutrophils and histiocytes. In NC38 (necropsy day 28, rank 5, moderate) lymphocytic aggregates are scattered within the alveolar septa and predominately mixed with histiocytes and rare neutrophils.

**NKG2a$^+$ cells and PD-1$^+$ T cells decrease in blood and increase in BAL overtime.** Infiltration of lymphocytes was observed in the lungs of SARS-CoV-2 infected AGM and RM at necropsy and was more robust in animals with severe disease (Fig. 5a). We first analyzed the kinetics of innate NK cells subsets, that we defined as HLA-DR$^-$ CD3$^-$ cells expressing the NKG2a receptors, which are known to traffic to the lung[45] (Fig. 5b–f and Supplementary Fig. 8). Interestingly, the frequencies of NKG2a$^+$ cells significantly decreased in peripheral blood of both AGM and RM over the course of infection (baseline vs. week 3, $P = 0.03$) (Fig. 5b). Consistent with the pattern described for the myeloid population, we also observed a concomitant increase in the frequencies of NKG2a$^+$ cells in BAL (baseline vs. week 3, $P = 0.05$) (Fig. 5c, d and Supplementary Fig. 8a–c). Four out of four AGM had decreased NKG2a$^+$ cells in peripheral blood, coinciding with an increase of the same cells in BAL. While the pattern of NKG2a$^+$ cells in peripheral blood of RM was not statistically significant, nonetheless in this species there was a marked increase of those cells in the BAL. These results suggest that the loss of NKG2a$^+$ cells in the PBMCs and the gain of NKG2a$^+$ cells in the BAL may coincide with the egress of these cells from peripheral blood to tissues. Accordingly, the frequency of NKG2a$^+$ cells expressing CXCR3, a marker associated with NK cell migration to the lung[46] also significantly decreased in blood over time (baseline vs. week 3, $P = 0.03$, Fig. 5e). Despite this change in the lymphocyte phenotype, the absolute number of lymphocytes did not change over the first 2 weeks of infection

(Fig. 5f) however it is important to note that all three remaining AGM had an increase at week 3 (Supplementary Fig. 8d, e).

tSNE analysis revealed an increase in the level of PD-1 at weeks 2 and 3 post-infection, overlapping with spatial regions corresponding to both CD4$^+$ T and CD8$^+$ T cells (Fig. 5g and Supplementary Fig. 9a). In fact, the frequency of PD-1$^+$ CD4$^+$ T cells was elevated in the blood of both RM and AGM (Fig. 5h, i, respectively). During the first week following the infection we observed an increase in IP-10 and IFN-γ (Supplementary Fig. 9b), however, we also observed a switch toward Th2-type responses characterized by the increase of Th2 cytokines IL-5 and IL-13 over time (Supplementary Fig. 9b). Accordingly, a gradual increase of Th2-type cells (defined as CXCR3$^-$ CCR6$^-$ CD4$^+$ T cells)[24] was seen in the blood within the PD-1 positive population (Fig. 5j). The increase of PD-1$^+$ Th2 cells was significantly and positively associated with IL-4 cytokine levels ($P = 0.016$, $R = 0.94$, Spearman test) (Supplementary Fig. 9c). Moreover, we observed an increased level of PD-1$^+$ CD8$^+$ T cells in BAL (baseline vs. week 2, $P = 0.015$; baseline vs. week 3, $P = 0.03$) (Fig. 5k). Animals divided by species are shown in Supplementary Fig. 9d, e. Associations between viral replication and the frequency of PD-1$^+$ CD4$^+$ T cells in the blood and PD-1$^+$ CD8$^+$ T cells in BAL at week 3 were found (Supplementary Fig. 9g–i), however, they were most likely driven by the AGM but not RM which had no detectable viral replication in bronchial brushes at that specific time point.

The ratio between the anti-inflammatory and pro-inflammatory responses is associated with disease progression. The pro-inflammatory cytokine IL-6 has been described as a strong predictor to disease progression in humans[3,7]. We measured IL-6 and IL-10 in the plasma of all the animals (Fig. 6a). While IL-6 levels alone did not associate with disease severity in NHPs, the ratio of IL-10–IL-6 levels measured in

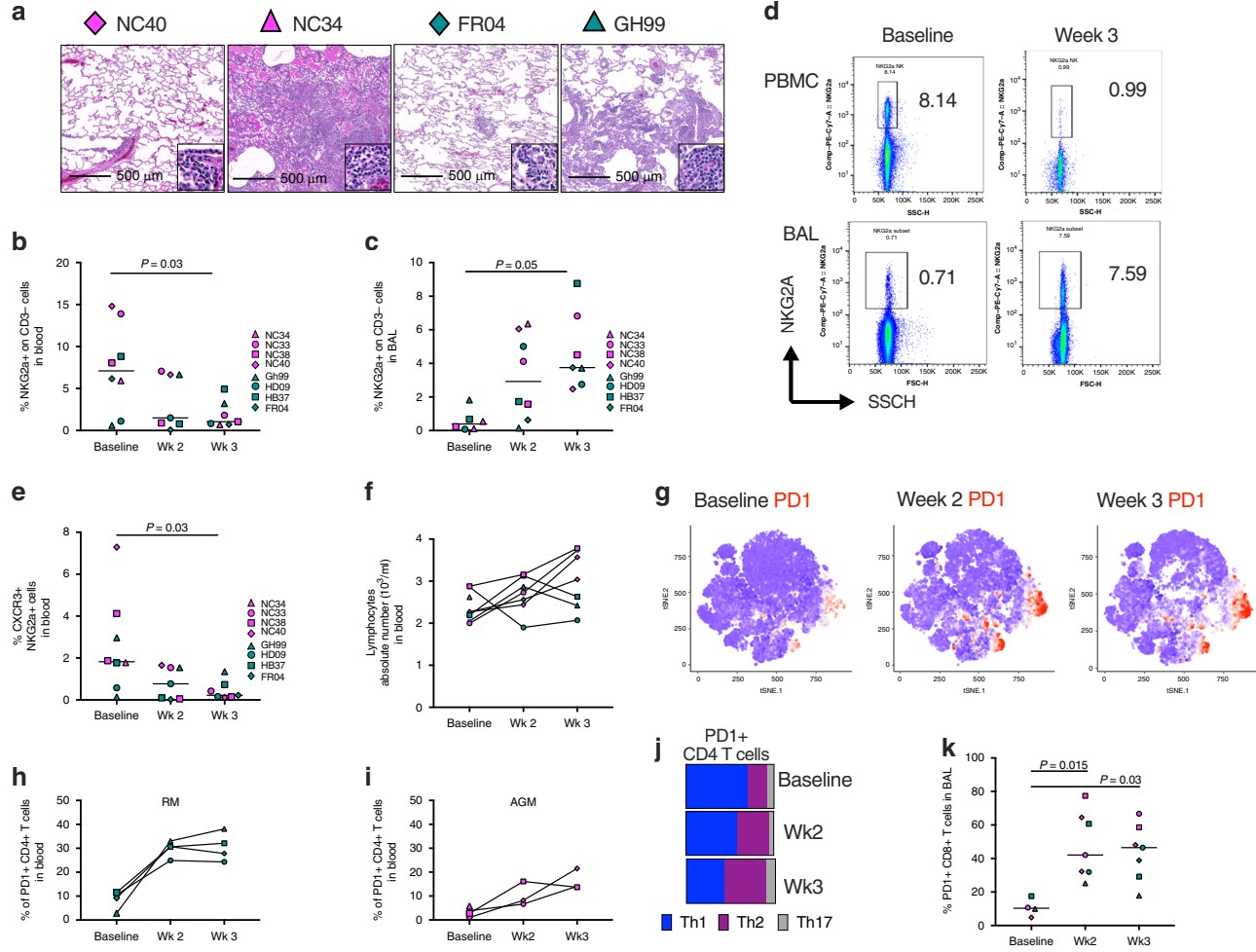

**Fig. 5 Lymphocytes infiltration in the lungs is associated with disease severity. a** Histopathologic findings showing infiltration of lymphocytes in the lung of AGM (NC40 and NC34) and RM (FR04 and GH99) with mild (NC40 and FR04), moderate (GH99), and severe disease (NC34). **b** Frequency of NKG2a$^+$ cells in the blood or **c** BAL of 4 AGM and 4 RM at baseline, 2 weeks and 3 weeks post SARS-CoV-2 infection by flow cytometry. **d** Representative flow cytometry plot showing a decreased frequency of NGK2a$^+$ cells in PBMCs and increased frequency in BAL to baseline in at week 3 post-SARS-CoV-2 infection respect to baseline. One animal for each species is shown. **e** Decrease in the frequency of CXCR3$^+$ NKG2a$^+$ cells in the blood at 2 weeks and 3 weeks post SARS-CoV-2 infection compared to the baseline. **f** Absolute number of lymphocytes before and after infection. **g** tSNE plots displaying kinetics of PD-1 expression (in red) on lymphocytic cell populations over time. **h** Frequency of PD-1$^+$ Th2 (CXCR3$^-$ CCR6$^-$) CD4$^+$ T cells is increased in RM and **i** in AGM overtime following the infection. **j** Changes in the percentage of Th2 (purple), Th1 (blue) and Th17 (gray) cells within the total PD-1$^+$ CD4$^+$ T cell population. **k** Significant increase in the percentage of PD-1$^+$ CD8$^+$ T cells in BAL. Analysis of variance (ANOVA) was used for comparing statistical differences between multiple groups, followed by Dunn's multiple comparison tests for **b**, **c**, and **k**.

plasma associated with both the ranking of disease severity and with the pathological score that was obtained by analyzing the extension of the inflammation and edema of the lungs ($R = 0.76$ $P = 0.037$; and $R = -0.76$, $P = 0.03$, by the Spearman correlation test, all animals) (Fig. 6b, c). To investigate further, we also measured the levels of tryptophan and kynurenine in plasma at baseline, week 1, week 3, and at necropsy (Supplementary Fig. 10). Kynurenine was elevated and significantly increased at week 3 compared to baseline levels ($P = 0.013$ adj.). The kynurenine (Kyn)/ tryptophan (Tryp) ratio has been used as a measurement of the indoleamine 2,3-dioxygenase (IDO) activity and is often associated with T regulatory cell function[21]. In line with the observed association for the IL-10:IL-6 ratio, the Kyn/ Tryp ratio was also negatively correlated with the pathology score (1 = mild–18 = severe) suggesting that animals with higher levels of immune suppression had less inflammation in the lung and better disease outcome (Fig. 6d).

These findings show a complex series of immune events in the lung that are most likely initiated as a host response to the virus

and then evolving over time toward less antiviral responses, as summarized in Fig. 6e.

## Discussion

There is an urgent need to develop safe and effective strategies to prevent and treat COVID-19. The development of such strategies requires a clear understanding of the immune cells involved in COVID-19 and their unique role in pathogenesis. Clinical observations in patients and few studies on non-human primates have started to shed light on the immune aspects involved in COVID-19[8,9,11,22,47,48]. Here, we further characterized immune events following SARS-CoV-2 infection in two NHPs models recapitulating crucial aspects of COVID-19 lung disease[16,18].

Altogether our findings provide the first evidence in both AGM and RM of a multiphasic immune response characterized by (1) an initial inflammatory phase with monocyte recruitment to the lung (2) a gradual switch from a type 1 to type 2 response, and (3) a "make it or break it" phase with either an increase in anti-inflammatory

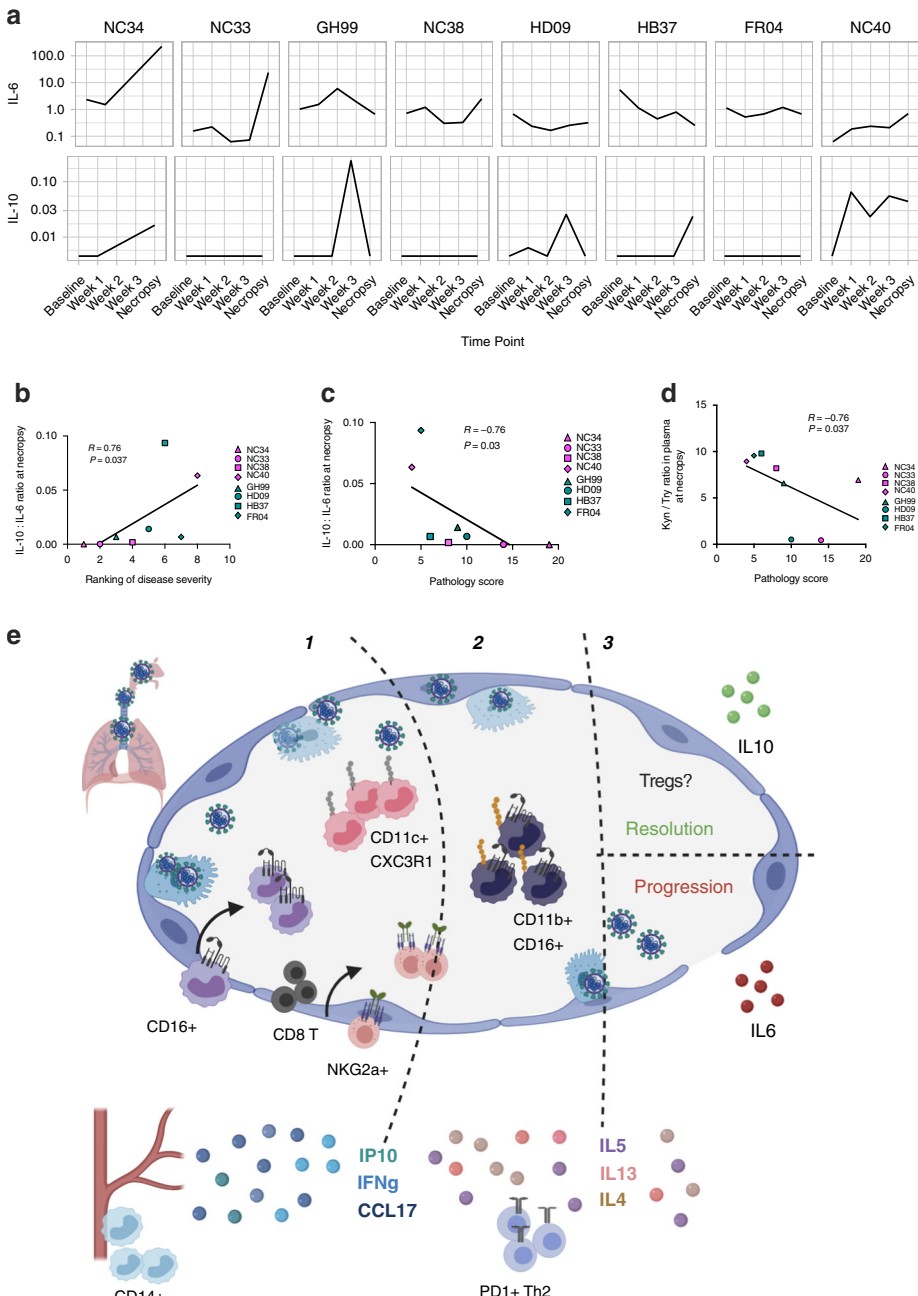

**Fig. 6 Increased IL-6 in the absence of IL-10 is associated with disease progression.** Levels of IL-6 and IL-10 in plasma over time (pg/mL) (**a**) and **b** correlation between the ratio of IL-10 to IL-6 levels measured in plasma and the ranking of disease severity or **c** with pathology score (n = 8), by the Spearman correlation test. **d** Association between Kyn/Tryp ratio in plasma (pg/mL) and pathology score (n = 8, Spearman test). **e** Summary of immune events in the lungs and blood of SARS-CoV-2 infected animals, from the 1-acute phase 2-switch to Th2-type responses and 3-resolution phase.

cytokines IL-10 and regulatory cell subsets with suppressive activity or IL-6, a pro-inflammatory cytokine associated with disease progression in humans, as summarized in Fig. 6e.

By examining the temporal and spatial distribution of monocytes, myelomonocytic cells, and macrophages in blood and BAL and lungs, we confirmed and extended on studies in humans and NHPs suggesting increased myelopoiesis with the recruitment of myeloid cells from the blood to lungs during the infection that is initially driven by the virus replicating in the lung[3,16,22]. Similar to what has been described by Wilk et al. and Thevarajan et al. in humans, we observed an increase of classical CD14[hi] monocytes and a corresponding drop in the frequency of monocytes expressing CD16 in the peripheral blood[22,23]. Importantly, we

extend on these findings by showing a simultaneous increase in the frequency of patrolling CD16[hi] HLA-DR[+] monocytes in BAL occurring in both species and by describing the kinetics of two CD16[+] HLA-DR[+] CD206[−] macrophages populations that preferentially populate the alveolar interstitium and exhibit features of antigen-presenting cells[49]. CD11c[+] CD16[+] cells arising from intravascular CD16[hi] patrolling monocytes in the acute phase and associating with IL-6 levels, and a CD11b[+] macrophage population known to be involved in the resolution of inflammation in the lung[20,33,39,40]. Higher frequencies of CD11b[+] monocytes with increased disease severity has been observed in the blood of humans[8]. The origin and the exact function of these interstitial macrophages remain to be determined.

Differently than what was observed in humans and previously reported data in RM, we did not detect an increase in the number of neutrophils, as these responses may have come up earlier than 1 week after infection as shown by others[16]. At necropsy, CD11b[+] MPO[+] neutrophils were present in the lungs of animals with severe disease as observed in humans. More studies are needed to determine if SARS-CoV-2 results in the accumulation of immature neutrophils as seen in humans[8,9].

Common to both species was the observation that animals with higher levels of myeloid and lymphoid cell infiltration in the lung had worse disease outcomes. Several studies in both SARS-CoV-1 and CoV-2 have reported depleted peripheral NK cell counts in severe patients[11,22,25,47]. Our data support the notion of NKG2a[+] cell trafficking to the lung during SARS-CoV-2 infection, most likely mediated by IP-10 (CXCL-10), which promotes the recruitment of CXCR3[+] Th1 thymocytes and NKs[50,51]. The expression of NKG2a has been linked to both activation and inhibition of inflammation[52]; the function in SARS-CoV-2 remains to be investigated. In the periphery, we also observed increased levels of PD-1 on T cells. This is consistent with findings in humans describing an exhaustive T cell phenotype during SARS-CoV-1 and CoV-2 infection[11,53].

We identified TARC (CCL17) as a possible mediator of myeloid recruitment in the lung during SARS-CoV-2. Both dendritic cells and CD11b[+] macrophages in the lung have been identified as a source of CCL17 in the airways in other pulmonary inflammatory diseases[54,55]. These cells promote the recruitment of T helper type 2 (Th2) lymphocytes into the airways[54]. Interestingly, we observed a gradual increase in the recruitment of Th2 lymphocytes into the airway and of Th2-type cytokines such as IL-5 and IL-13. These findings show a complex series of immune events in the lung that are most likely initiated as a host response to the virus, and that they may evolve into other less effective antiviral responses as either an attempt to tissue repair/resolution of inflammation, as summarized in Fig. 6e or as the result of virus direct immunomodulatory effects on host responses[6].

Intrinsic species-specific susceptibility to disease progression has been described for NHP in SARS-CoV-1[6] with AGM being more susceptible than rhesus to severe disease. There were unexpected similarities in the kinetics and the quality of the immune responses analyzed in RM and AGM, particularly during the acute phase of infection, however, AGM did not clear the infection, as well as RM and two AGM, had the most severe symptoms[56]. While the number of animals enrolled in this study was not sufficient to look at species-specific differences and their effect on disease progression, we found that a higher ratio of IL-10: IL-6 in plasma was associated with disease severity. Undoubtedly, more studies are needed to prove or disprove the single contribution of IL-10 and IL-6 in the outcome of COVID-19 disease and to identify the source of these two potential key players[57]. The notion that dynamically balanced pro and anti-immune responses to SASR-CoV-2 have an important role in disease progression and outcome is supported by the association of the pathological score and the Kyn/Tryp suggesting a role of T regulatory cells in resolution of inflammation of the lungs in COVID-19.

Overall, this study uncovers critical steps in the immune events of the host response to SARS-CoV-2 and describes the correlates of viral replication and disease progression. We show viral driven inflammation/cell recruitment to the lung occurs rapidly, and lasts up to 4 weeks post-infection, which may explain in part the observed slow recovery from COVID-19 seen humans. Finally, these findings have the potential to aid the interpretation of the results of the human trials that are currently underway for several vaccine candidates and may be instrumental in generating targeted therapeutic strategies aimed at resolving the severe immune deregulation of the lung of COVID-19 patients.

## Methods

**Ethical statement on animal use and SARS-CoV-2 handling**. Animals and infection. The Institutional Animal Care and Use Committee of Tulane University reviewed and approved all the procedures for this study. The Tulane National Primate Research Center is fully accredited by the AAALAC. All animals were cared for in accordance with the ILAR Guide for the Care and Use of Laboratory Animals 8th Edition. The Tulane University Institutional Biosafety Committee approved the procedures for sample handling, inactivation, and transfer from BSL3 containment.

**Animals and infection**. Four adult-aged African green monkeys (Caribbean origin) and four rhesus macaques (*Macaca mulatta*, Indian ancestry) were exposed to SARS-CoV-2; 2019-nCoV/USA-WA1/2020409 (MN985325.1). The virus stock was prepared in Vero E6 cells and the sequence confirmed by PCR and/or Sanger sequencing. Plaque assays were performed in Vero E6 cells. The rhesus macaques were from the Tulane National Primate Research Center breeding colony pathogen-free for simian type D retrovirus (SRV), *macacine herpesvirus 1* (B virus), *simian immunodeficiency virus* (SIV), *simian T cell lymphotropic/leukemia virus* (STLV), *measles virus* (MV) and *Mycobacterium tuberculosis* (TB). The African green monkeys were wild-caught and were kept at the TNPRC for over a year before being assigned to this study. To mimic different possible routes of infection in humans, animals were exposed to the virus either by aerosol (inhaled dose of $2.0 \times 10^3$ and $2.5 \times 10^3$ TCID$_{50}$) or by inoculating a cumulative dose of $3.61 \times 10^6$ PFU through multiple routes (oral, nasal, intratracheal, conjunctival) (Supplementary Table 1).

The multi-route exposure was given to four animals, one adult RM male (GH99, 14 years old), one adult RM female (HD09, 13 years old) and two AGM, one aged male (NC40, 16 years old, approximately) and one aged female (NC33, 16 years old, approximately). An additional two adult males RM (FR04 and HB37, 15 and 13 years old, respectively) and one aged male and female AGM: NC34 and NC38 (16 years old, approximately) were exposed by aerosol (Supplementary Table 1). Rhesus macaques tend to live ~20 years in captivity, while African green monkeys live ~15 years[58]. The age of the macaques is generally multiplied by three when compared to humans, and GH99 had signs of aged lungs that were not associated with COVID-19.

**Cell count and CO$_2$ and SpO$_2$ pressure measurements**. Total white blood cell counts, white blood cell differentials, red blood cell counts, platelet counts, hematocrit values, total hemoglobin concentrations, mean cell volumes, mean corpuscular volumes, and mean corpuscular hemoglobin concentrations were analyzed from blood collected in tubes containing EDTA using an SYSMEX XN-1000v Hematology Analyzer in the Level 2 Lab and a Sysmex XS-1000 in the Level 3 Lab. CO$_2$ measurements were taken using the Piccolo Express Chemistry Analyzer in the Level 3 lab and are part of the Metabolic Comprehensive Panel. SpO$_2$ values were obtained using the Masimo Rad-57 Handheld Pulse Oximeter (Supplementary Table 3).

**Sample collection and processing**. Blood EDTA was collected by venipuncture and bronchial brush, and BAL by bronchoscopy. For the latter, the bronchoscope is introduced into the trachea and directed into one of the bronchi. The bronchoscope is advanced to the point where the diameter of the bronchoscope approximates the dimensions of the bronchus. The bronchial brush is passed through the channel of the bronchoscope. The brush is gently advanced until the lumen of the bronchus or bronchioles approximates the dimensions of the brush. The brush is then gently advanced back and forth against the mucosa to gather cells. The brush is removed from the channel of the bronchoscope and the brush is placed into 200 μL of DNA/RNA Shield X1 (Cat.# R1200, Zymo Research, Irvine, CA). The bronchoscope remains in place while two aliquots of 20 mL of saline each are instilled into the bronchus and subsequently aspirated. This procedure is then repeated in the opposite lung.

Blood was layered over Ficoll-Plaque Plus (GE Healthcare17144003) and centrifuged for 30 min in cap-locked adapters. Differential migration of cells during centrifugation results in the formation of layers containing different cell types and allows the collection of peripheral blood mononuclear cells (PBMCs) from other populations. PBMCs were collected, counted, and aliquoted in tubes or resuspended in freezing media containing Fetal Bovine Serum and DMSO. For bronchoalveolar lavage (BAL), cells and supernatant were separated by centrifugation (15 min at 2000 × g at RT) in cap-locked adapters, and cells were lysed in Ammonium Chloride Potassium (ACK) lysing buffer at room temperature for 5 min (Gibco# A1049201) washed in 2% FBS, and resuspended in PBS and counted under laminar flux hood in BCL3. Fresh cells were stained for flow cytometry.

**BAL and PBMCs staining**. Freshly obtained PBMCs and cells obtained from BAL were stained with appropriate antibody cocktails and incubated for 30 min at room temperature. Antibodies are listed in Supplementary Table 10. 3 μL of antibodies were used for all except for CD56, and PD-1 (4 μL). Cells were fixed with 4% or 2% PFA-BSA overnight. Samples were run on a FACSAria Fusion flow cytometer (Becton Dickinson) within a Biosafety Cabinet, and each sample was run for at

least 5 min. For all panels, the LIVE/DEAD Fixable Aqua Dead cell Stain kit was used (ThermoFisher# L34957). Monocytes subsets were gated as described in Supplementary Fig. 2a and as previously described[24]. Alveolar macrophages were gated based on the size and the expression of HLA-DR, CD206, and CD163 markers. Myeloid infiltrates were gated as cells negative for CD206 and CD163 and positive for HLA-DR, CD16, and CD11b. The gating strategy is shown in Supplementary Fig. 4g. NKG2a+ cells were gated on the CD3 negative parent population as shown (BAL FR04 week 2) in Supplementary Fig. 8a. Th1, Th2, and Th17 cells were gated as previously described[24]. tSNE analyses were performed in FlowJo using the opt-SNE algorithm and plots were created in R using the ggplot2 package[59].

**Histopathology and scoring.** Tissue samples were fixed in Z-fix (Anatech), embedded in paraffin, and 5-μm-thick sections were cut, adhered to charged glass slides, and stained with hematoxylin and eosin. All slides were scanned on a Zeiss Axio Scan.Z1 digital slide scanner. Images were acquired using HALO software (Indica Labs). Quantification of pathologic lesions was performed with HALO software. Machine learning (Random Forest) algorithms were trained by a board-certified pathologist to recognize fluidic and cellular inflammation in the lungs of the RM and AGM. Scores were assigned to fluidic and cellular inflammation, separately, based on the percentage of the lung affected. Fluidic and cellular inflammation scores were summated for all lobes and animals were ranked from 1 (highest sum) to 8 (lowest sum).

**Plasma cytokines and chemokines.** Plasma was collected by spinning and was thawed before use. Cytokines were measured using 497 Mesoscale Discovery. IFN-γ, IL-4, IL-6, IL-8, IL-10, and IL-13 were part of the V-Plex Pro-inflammatory Panel 1, 10-Plex. V-plex. The chemokine panel (#K15049D, Mesoscale Discovery, Rockville, Maryland) was also used, following the instructions of the kit. The plate was read on a MESO Quickplex 500 SQ120 machine.

Heatmaps were generated using the "pheatmap" package in R[60,61]. Data were normalized by dividing raw values at week 1 by baseline values for each animal, followed by the application of log₂. Values below the limit of detection were replaced with the lowest limit of detection value based on the standard curve for each run, or with the lowest value detected during the run, whichever was smaller. Bubble plots were generated using the "ggplot2" package in R, using the same normalized data shown in the heatmap[59]. Scatter plots were drawn using raw data points and display Pearson's correlation coefficients and a 95% confidence interval.

**RNA isolation.** Swab and bronchial brush samples were collected in 200 μL of DNA/RNA Shield X1 (Cat.# R1200, Zymo Research, Irvine, CA) and extracted for Viral RNA (vRNA) using the Quick-RNAViral kit (Cat.# R1034/5, Zymo Research). The Viral RNA Buffer dispensed directly to the swab in the DNA/RNA Shield. The swab was directly inserted into the spin column. The vRNA was eluted (45 μL) and 5 μL was added in a 0.1 mL fast 96-well optical microtiter plate (Cat.# 4346906, ThermoFisher, CA).

**Viral RNA detection.** RT-qPCR reaction TaqPath1-Step MultiplexMaster Mix was used (Cat.# A28527, ThermoFisher) along with the 2019-nCoV RUO Kit (Cat.# 10006713, IDTDNA, Coralville, IA) targeting the N1 amplicon of N gene of SARS2-nCoV19 (accession MN908947). The master mix was added to the microtiter plates covered with optical film (Cat.# 4311971; ThermoFisher) and then was vortexed and pulse centrifuged. The RT-qPCR program consisted of incubation at 25 °C for 2 min, RT incubation at 50 °C for 15 min, and an enzyme activation at 95 °C for 2 min followed by 40 cycles of denaturing step at 95 °C for 3 s and annealing at 60 °C for 30 s. Fluorescence signals were detected with an Applied Biosystems QuantStudio6 Sequence Detector. Data were captured and analyzed with Sequence Detector Software v1.3 (Applied Biosystems, Foster City, CA). Viral copy numbers were calculated by plotting Cq Values obtained from unknown (i.e. test) samples against a standard curve representing known viral copy numbers. The limit of detection of the assay was 10 copies per reaction volume. A 2019-nCoVpositive control (Cat.# 10006625, IDTDNA) were analyzed in parallel with every set of test samples. A non-template control (NTC) was also included.

**Immunohistochemistry.** Five μm sections of Formalin-fixed, paraffin-embedded lung were mounted on charged glass slides, baked overnight at 56 °C, and passed through Xylene, graded ethanol, and double distilled water to remove paraffin and rehydrate tissue sections. A microwave was used for heat-induced epitope retrieval, which was performed sequentially with high and low pH solutions from Vector Labs (H-3301 and H-3300). CD16+ and MPO+ cells were detected using MACH3 AP kits (Biocare Medical M3M532 or M3R533) and permanent red substrate (Dako K0640). MPO+ cells CD3+ cells CD11b+ cells CD68+/CD163+ cells, and CD206+ cells were detected using antibodies described in Supplementary Table 10. Slides were incubated with a blocking buffer comprised of either 1% normal donkey serum or 10% normal goat serum for 40 min. All primary antibodies were incubated for 60 min and all secondary antibodies for 40 min at room temperature, with thorough washing in between. DAPI (4′,6-diamidino-2-phenylindole) was used to label the nuclei of each section. Slides were mounted using a homemade

anti-quenching mounting media containing Mowiol (Calbiochem# 475904) and DABCO (Sigma# D2522) and imaged with a Zeiss Axio Slide Scanner.

**Arginase activity.** Arginase activity was analyzed on Plasma using the Arginase Activity Assay Kit (MAK112, Sigma-Aldrich, St. Louis, MO) following the manufacturer instructions. Briefly, samples were thawed on ice and, in order to deplete the urea, 40 μL of plasma were loaded in an Amicon®Ultra 10 K centrifugal filter (UFC501096 EMD Millipore), diluted with pure water to 500 μL, and centrifuged at 13,000 × g for 30 min at 4 °C. Following centrifugation, the eluted solution was discarded. Filtered samples were then diluted with pure water to 500 μL, and centrifuged at 13,000 × g for 30 min at 4 °C. At the end of centrifugation, the remaining volume of each sample was measured, and ultra-pure water was added to reach a final volume of 40 μL. Each sample was loaded into two wells of a 96-well plate (20 μL/well), representing the sample well, and the sample blank well, and 20 μL/well of ultra-pure water were added to each well. Together with samples, the plate was loaded with urea standard and water as positive and negative controls, respectively. Samples were loaded in singlicate, whereas controls were loaded in duplicate. Ten microliters of 5× substrate buffer, composed of Arginine Buffer and Mn Solution, were added to the wells except for sample blank wells, and they were incubated for 120 min at 37 °C. Following the incubation, 200 μL of Urea Reagent, composed of Reagents A and B, was added to each well to stop the reaction. Finally, 10 μL of 5× Substrate Buffer was added to the sample blank wells to have the same reagents proportion of reagents as the Sample Wells. After mixing, the plate was incubated for 60 min at RT and finally acquired with microplate spectrophotometer Power Wave XS2 (BioTek Instruments, Winooski, VT) to measure the absorbance at 430 nm ($A_{430}$) of each well. The arginase activity was determined per the following equation: $[(\text{sample well}) - (\text{blank})/(\text{urea} - \text{blank})] \times [(1 \text{ mM} \times 50 \times 103)/(40 \text{ μL} \times 120 \text{ min})]$, according to the manufacture instructions.

**Kynurenine and tryptophan plasma levels.** Tryptophan and Kynurenine plasma concentrations were measured by using the Tryptophan ELISA (Rocky Mountain Diagnostics, Colorado Springs, CO, USA, Cat.# BA E-2700) and Kynurenine ELISA commercial kits (Rocky Mountain Diagnostics, Colorado Springs, CO, USA, Cat.# BA E-2200). For tryptophan measurement, 20 μL of plasma was precipitated, the recovered supernatants were derivatized, and the product was used to perform the ELISA according to manufacturer instructions. For kynurenine assay, 10 μL of plasma were acylated and used to perform the ELISA according to manufacturer instructions. The data are presented as the ratio between kynurenine and tryptophan (Krn/Try) levels.

**Statistical analysis.** Correlation analyses were performed using the non-parametric Spearman rank correlation method (two-tailed, 95% confidence) using exact permutation P-values when ranks were used, or Pearson's correlation test when both variables were continuous. Analysis of variance (ANOVA) was used for comparing statistical differences between multiple groups, followed by Dunn's multiple comparison tests. Kruskal–Wallis ANOVA or Mann–Whitney tests were used. All statistical analyses were performed using GraphPad Prism (version 8.4.2 GraphPad Software, La Jolla California USA) and R software (URL: http://www.R-project.org/). Correlation plots were created using the "Performance Analytics" and "corrr" plots, respectively, in R[61–63].

**Statistics and reproducibility.** All immunohistochemical techniques, performed on the lung tissues, have been validated and approved for use by the center's Unit of Quality Assurance. Rigid standard operating procedure is followed by experienced core personnel.

**Reporting summary.** Further information on research design is available in the Nature Research Reporting Summary linked to this article.

## Data availability
The data supporting the findings of this study are available within the article and its Supplementary Information files, or are available from the authors upon request. The raw data supporting the findings that are not in the Tables can be accessed found here: https://doi.org/10.6084/m9.figshare.13148597.v1. Source data are provided with this paper.

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

## Acknowledgements
We would like to thank Dr. Joseph (JC) Mudd and Dr. Cariappa Annaiah for critical reading of the manuscript and Dr. Amitinder (Miti) Kaur, Megan Varnado, and Kaitlin Didier from the TNPRC Flow Cytometry Core for their help. We thank Angela Birnbaum for reviewing and optimizing all technical SOPs and overseeing the safety of this study. We would also like to thank Fast Grant Funding for COVID-19 Science for partially funding this work and the NIH for supporting this work through the TNPRC base grant (P51 OD011104 59).

## Author contributions
M.D.F. analyzed data composed figures and wrote the manuscript, B.R.V. was the lead pathologist and L.A.D.-M. was the project veterinarian, contributed to study design, and writing of the I.A.C.U.C. and manuscript. C.C.M. designed I.H.C. panels and performed the I.H.C. staining, G.Z. wrote the manuscript, K.E.R.-L. was a project veterinarian, made clinical assessments and collected samples, C.J.M. processed and analyzed samples for RT-qPCR, contributed; E.H.H. provided the mesoscale data, T.P.P. stained the samples with G.L. and B.M.T. G.L. contributed to study design, provided administrative support, and aided with sample processing and archiving. N.G. contributed to study design, study coordination, sample processing, and S.O.P. development. P.K.D. processed and analyzed viral load data. C.J.R. conceived and performed aerosol experiments. R.P.B. contributed to study design, analysis of clinical and imaging results, and writing the manuscripts. N.J.M. performed staining of B.A.L. and contributed to writing the manuscript. T.F. contributed to the animal study design and planning and contributed to writing the manuscript. J.R. designed and supported the animal study and helped with the writing of the manuscript. M.V. conceived and supported the study, run experiments, analyzed data, and wrote the manuscript.

## Competing interests
The authors declare no competing interests.
