## [Peer Review File · Nature Communications]

Reviewers' Comments:

Reviewer #1:

Remarks to the Author:

Here the authors studied immune responses and pathology during acute SARS-CoV-2 infection of nonhuman primates (NHP). They found changes to bronchial monocyte populations were associated with viral load and IL-6, and alterations to monocyte populations in the lung. Interestingly, macrophages were accumulating and long-lasting in all Covid animals despite disease severity, but IL-10:IL-6 ratios was correlated with less severe disease. Overall the manuscript is highly important for understanding mechanisms of Covid disease severity and demonstrate the importance of NHP as a Covid model. Minor issues below.

1. The abstract would benefit from a more substantial conclusion regarding the macrophages. What do these macrophage changes mean for pathology and disease? Why did they not include more of the overall results such as the CCL17?
2. Macrophage gating is unclear. Was CD14 used to identify the macrophages? They demonstrate gating in figure 1 but should make the figures more clear as to what each population is and how it was defined (ie classical monocytes CD14++...)
3. The IHC figures could all be better labeled so one can look at them and immediately understand what each stain/color is and doesn't have to reference the figure legends.
4. The IL-10 is very interesting and should be highlighted. Did the authors assess Tregs at all? They should at least discuss this as a possibility if it was not done originally.

Reviewer #2:

Remarks to the Author:

Comments to authors: Here the authors have studied frequencies and phenotypes of myeloid cells in blood and lung of SARS-CoV-2 infected NHPs. The authors find elevated levels of proinflammatory monocytes which they argue is associated with disease severity. Overall the manuscript is interesting. However, there are several confounding variables that the authors need to control.

1. There were only 8 animals infected. The animals were of different species, different genders, different ages, and were infected via different routes. It is inappropriate to group them all together. They should be grouped according to their infection routes, species, age, and gender. Even with these, much of the statistical variation is attributed to the outlier animal (GH99).
2. The authors should show longitudinal pulse oxygen levels and temperatures for all of the animals.
3. The authors should provide representative flow gating. It's possible there are contaminating NK cells in their 14 vs 16 plots. Moreover, myeloid cells from the lung (BAL in particular) are notoriously difficult to study by flow cytometry.
4. Why are different numbers available for week 1 timepoints for the phenotypic comparisons, but not for the volumetric counts (Figure 1)?
5. Why are data from NC34 available from week 2 for VL (1B), but no other data?
6. The utility of the IL10/IL6 ratio is unclear given many of the animals had undetected IL10 (5/8 at necropsy to this reviewer's eyes). There's clearly a problem with the AGM/RM symbols in figure 5B/C also. (in B three AGM have IL10/IL6 ratios of 0, but in C one AGM and 2 rhesus have IL10/IL6 ratios of 0).

Reviewer #3:

Remarks to the Author:

In the given manuscript, Fahlberg et al. present convincing evidence regarding the dynamics of distinct immune cells within the lungs and blood of 2 non-human primate (NHPs) models namely

African green monkeys (AGM) and Rhesus macaques (RM) at distinct time points of the SARS-CoV2 infection. This is a well-controlled, statistically-sound, fascinating and significant study and is a very well written manuscript which recapitulates a lot of findings recently reported for human COVID-19 patients in NHPs. The manuscripts are Schulte-Schrepping et al. Cell 2020 (doi:<https://doi.org/10.1016/j.cell.2020.08.001>) and Silvin et al. Cell 2020 (doi:<https://doi.org/10.1016/j.cell.2020.08.002>).

Having said that the three pieces of novelty that this manuscript brings to the community is (i) the inclusion of exhaustive time points in the study, (ii) the identification of CCL17 as a potential mediator of myeloid recruitment to the lungs during COVID-19 and (iii) potential use of IL-10 to IL-6 ratios in predicting outcomes and course of COVID-19 disease; all these studies being done in AGMs and RMs that have previously been shown to be good models of SARS-CoV2 infection by the authors and others. Given the recent precedence for their observations in human studies, there are several questions that should be addressed for a more comprehensive presentation of this study. Also considering the expenses associated with doing new NHP studies and the COVID-19 related significance of this work, suggested concerns have been carefully chosen to be address-able with the existing datasets and samples. The comments in order of their mention in the manuscript are:

1. Line 39: "...and validate NHP as models to test immune therapies" is inaccurate. The results do not validate NHPs as models to test immune therapies but suggest that they could make good models for the same. Validation of therapies will need a lot larger sample size and use of candidate therapies both of which were not within the scope of this study.

2. In Figure 1, it would be interesting to know if and how the absolute numbers of classical and non-classical monocytes correlate with viral titers at the tested timepoints.

3. Along same lines, in Figure 1, How do the absolute numbers of classical and non-classical monocytes correlate with the disease severity? In Figure 1H and 1I, HD09 and GH99 have the least non-classical monocytes and most classical monocyte numbers and these also happen to be the RMs with the worst disease severity (Table 1). This could suggest that non-classical monocytes are reflective of worse outcomes which is in lines with the published human data. This point 2 also makes it inaccurate for the conclusion made on Line 150: "Pre-infection levels were restored at 3 weeks post-infection (week 1 vs. week 3: classical $P = 0.03$ and non-classical $P = 0.03$). Clearly HD09 and GH99 (the RMs with worse outcomes) never regained their baseline numbers of non-classical monocytes.

These reasons make it necessary to study the cellular effects that the authors have reported in with stratification of NHPs with worse vs milder outcomes, albeit in supplemental data with a smaller sample size. It would be a phenomenal addition to the already existing dataset.

4. In Supplemental Figure SF 1G, the authors did not find any changes in neutrophil numbers within the blood however they do not mention if and what changes they saw for neutrophils in the BAL samples. It would be helpful to show if and how the absolute numbers of BAL neutrophils correlate with viral titers and disease severity at the tested timepoints.

5. Line 151-152: This statement might need toning down since the authors present no direct evidence of these monocytes having anti-viral activity. It is implied but not proved.

6. Human studies show that patients with severe COVID-19 exhibit emergency myelopoiesis which is visible as presence of CD11b^{low/-} neutrophils (Schulte-Schrepping et al. Cell 2020 and Silvin et al. Cell 2020). How did the surface CD11b levels look like for neutrophils (both BAL and blood) over the duration of infection? It would be great to show that the NHP models reflect similar changes as in humans, given that the current panel already has CD11b? If anything, it would add strength to the manuscript.

7. Supplemental Figure SF 2I is also difficult to understand. A more detailed legend explaining how the PCA was plotted will help.
8. Supplemental table 3 is difficult to understand. A more detailed legend will help. I am not sure what PC1-PC8 stand for? If these are principal components, how do they reflect what animal is what PC? Providing raw baseline values of each cytokine would be easier to comprehend.
9. In Figure 1K: I am not sure how to read Figure 1K. There are 4 AGMs or RMs but a lot more dots. I assume each dot stands for chemokine levels in one animal? In that case maybe giving different shape for each chemokine with species specific color will help?
10. Line 167: "Overall this analysis revealed that animals with the most severe disease outcomes had higher levels of chemokines at week 1, including IP-10, TARC (CCL17) (Fig. 1L)". Does this stand true when AGM and RM analyzed separately considering baseline difference for the NHPs?
11. Figure 1L: Can the data for weeks 2,3,4 also be provided as in Fig 1L as supplemental data? It will help the readers decide the strength of the conclusions.
12. In lines 185-199, the authors use the term macrophages to categorize the distinct myelomonocytic cells without use of markers that can discriminate between monocytes or macrophages. Monocytes also express CD11b and CD16 and considering that monocyte to macrophage transition is a continuum, it would be inaccurate to call these cells macrophages. I would suggest use of monocyte-derived cells or myeloid cells or myelomonocytic cells.
13. Human studies show that patients with severe COVID-19 possess HLA-DR^{low}CD163^{hi} monocytes in the blood. How did surface CD163 levels change in macrophage subsets represented in Fig 2G-I at different timepoints?
14. In Figure 3A. Moving the animal ID# off from the middle of the H&E images will make the image more visually appealing.
15. Figure 4: How do the frequency of NK cells, CD4 T cells, CD8 T cells, PD-1+ CD4 T cells, PD-1+ CD8 T cells correlate with viral burdens and disease severity.
16. It will be useful to know where the distinct lymphocyte cells aggregate in the lungs of SARS-CoV2 infected NHPs. Do they intermingle with certain specific type of myeloid cells? Do they exist in certain parts of the lungs?
17. Figure 4G: Not sure if the tSNE accurately depicts the CD4 and CD8 regions. It makes comprehending the conclusion on Line 250 difficult.
18. Reference #14 doesn't really help the way it has been listed without title/journal/year.
19. Line 319:" For the first time, we identified TARC (CCL17) as central to myeloid recruitment in the lung during SARS-CoV-2." is inaccurate. There is no direct evidence (only correlation) for this in this work. This statement needs to be toned down as suggestion and not causative evidence.
21. How does the age of the monkeys included in these studies compare with age of humans? COVID-19 has worse outcomes in older human patients. A discussion on this would be helpful.

Point-by-point response to the reviewers

We would like to thank the reviewers for the great suggestions. We have addressed all the points as described below, and we have marked all the changes in the current manuscript in red.

Reviewer #1 (Remarks to the Author):

1. The abstract would benefit from a more substantial conclusion regarding the macrophages. What do these macrophage changes mean for pathology and disease? Why did they not include more of the overall results such as the CCL17?

We have now modified the abstract accordingly.

2. Macrophage gating is unclear. Was CD14 used to identify the macrophages? They demonstrate gating in figure 1 but should make the figures more clear as to what each population is and how it was defined (ie classical monocytes CD14++...)

We have included our gating strategy (Supplementary Figure 2A) and modified the nomenclature accordingly.

3. The IHC figures could all be better labeled so one can look at them and immediately understand what each stain/color is and doesn't have to reference the figure legends.

We have modified Figure 3 as suggested.

4. The IL-10 is very interesting and should be highlighted. Did the authors assess Tregs at all? They should at least discuss this as a possibility if it was not done originally.

We have now included the analysis for Arginase 1 (myeloid derived suppressor cells) and the tryptophan kynurenine ratio as a functional measure of T reg (IDO) activity (Figure 6D and Supplementary Fig 10) and expanded the discussion accordingly.

Reviewer #2 (Remarks to the Author):

Comments to authors: Here the authors have studied frequencies and phenotypes of myeloid cells in blood and lung of SARS-CoV-2 infected NHPs. The authors find elevated levels of proinflammatory monocytes which they argue is associated with disease severity. Overall the manuscript is interesting. However, there are several confounding variables that the authors need to control.

1. There were only 8 animals infected. The animals were of different species, different genders, different ages, and were infected via different routes. It is inappropriate to group them all together. They should be grouped according to their infection routes, species, age, and gender.

In our original submission we have been very careful to separate the AGM and RM by showing them in different colors and also by showing and analyzing them separately in the Supplementary Figures. We have now provided additional statistical analysis showing that no differences in disease severity nor viral replication could be found between females and males nor challenge routes (Supplementary Table 4). We think the range of the animal age (13-16) is not large enough that we can stratify accordingly.

Even with these, much of the statistical variation is attributed to the outlier animal (GH99). We have now provided evidence that GH99 is not driving the main conclusions of our analysis by including additional figures and statistics in Supplementary Figures 1G, 1M, and 3K where this animal was removed. Please note that all the analysis retained significance when this animal is excluded.

2. The authors should show longitudinal pulse oxygen levels and temperatures for all of the animals. We thank the reviewer for the suggestion. Information can now be found in Supplementary Table 3 (lines 130 -134).

3. The authors should provide representative flow gating. It's possible there are contaminating NK cells in their 14 vs 16 plots. Moreover, myeloid cells from the lung (BAL in particular) are notoriously difficult to study by flow cytometry. We have included gating strategy as suggested by the reviewer (Supplementary Figure 2A and Figure 7A). Myelomonocytic cells gated in BAL were originally shown in Supplementary Figure 3 which is now SF4.

4. are different numbers available for week 1 timepoints for the phenotypic comparisons, but not for the volumetric counts (Figure 1)?

We have now provided absolute counts for the subsets in Supplementary Figure 2. We have provided numbers of animals in figure legends.

5. Why are data from NC34 available from week 2 for VL (1B), but no other data?

NC34 was euthanized on week 1, day 8 because of respiratory distress as stated at line 117-118.

6. The utility of the IL10/IL6 ratio is unclear given many of the animals had undetected IL10 (5/8 at necropsy to this reviewer's eyes). There's clearly a problem with the AGM/RM symbols in figure 5B/C also. (in B three AGM have IL10/IL6 ratios of 0, but in C one AGM and 2 rhesus have IL10/IL6 ratios of 0).

We agree more studies are needed to understand the utility of the IL-10 /IL-6 ratio as we have stated in the discussion, we have corrected the symbols and per request by the other two reviewers we now have elaborated on this point by adding ulterior information regarding the possible role of T regulatory cells /myeloid derived suppressor cells (reviewer1) and added analysis of Tryptophan catabolism as a measurement for IDO/treg activity (Figure 6D supplementary Figure 10).

Reviewer #3 (Remarks to the Author):

In the given manuscript, Fahlberg et al. present convincing evidence regarding the dynamics of distinct immune cells within the lungs and blood of 2 non-human primate (NHPs) models namely African green monkeys (AGM) and Rhesus macaques (RM) at distinct time points of the SARS-CoV2 infection. This is a well-controlled, statistically-sound, fascinating and significant study and is a very well written manuscript which recapitulates a lot of findings recently reported for human COVID-19 patients in NHPs.

The manuscripts are Schulte-Schrepping et al. Cell 2020 (doi:<https://doi.org/10.1016/j.cell.2020.08.001>) and Silvin et al. Cell 2020 (doi:<https://doi.org/10.1016/j.cell.2020.08.002>).

Having said that the three pieces of novelty that this manuscript brings to the community is (i) the inclusion of exhaustive time points in the study, (ii) the identification of CCL17 as a potential mediator of myeloid recruitment to the lungs during COVID-19 and (iii) potential use of IL-10 to IL-6 ratios in predicting outcomes and course of COVID-19 disease; all these studies being done in AGMs and RMs that have previously been shown to be good models of SARS-CoV2 infection by the authors and others.

Given the recent precedence for their observations in human studies, there are several questions that should be addressed for a more comprehensive presentation of this study. Also considering the expenses associated with doing new NHP studies and the COVID-19 related significance of this work, suggested concerns have been carefully chosen to be address-able with the existing datasets and samples.

The comments in order of their mention in the manuscript are:

1. Line 39: "...and validate NHP as models to test immune therapies" is inaccurate. The results do not validate NHPs as models to test immune therapies but suggest that they could make good models for the same. Validation of therapies will need a lot larger sample size and use of candidate therapies both of which were not within the scope of this study. **We agree with this reviewer and we have modified the text accordingly in Abstract and Introduction (last paragraph).**

2. In Figure 1, it would be interesting to know if and how the absolute numbers of classical and non-classical monocytes correlate with viral titers at the tested timepoints.

3. Along the same lines, in Figure 1, How do the absolute numbers of classical and non-classical monocytes correlate with the disease severity?

In Figure 1H and 1I, HD09 and GH99 have the least non-classical monocytes and most classical monocyte numbers and these also happen to be the RMs with the worst disease severity (Table 1). This could suggest that non-classical monocytes are reflective of worse outcomes which is in lines with the published human data.

This point 2 also makes it inaccurate for the conclusion made **on Line 150**: "Pre-infection levels were restored at 3 weeks post-infection (week 1 vs. week 3: classical $P = 0.03$ and non-classical $P = 0.03$). Clearly HD09 and GH99 (the RMs with worse outcomes) never regained their baseline numbers of non-classical monocytes.

These reasons make it necessary to study the cellular effects that the authors have reported in with stratification of NHPs with worse vs milder outcomes, albeit in supplemental data with a smaller sample size. It would be a phenomenal addition to the already existing dataset.

We thank the reviewer for these suggestions.

We have now:

1) included the absolute numbers for the monocytic subsets in Supplementary Figure 2H-I (and line 166-183)

2) Described associations with frequency and numbers and viral load /disease and add a new Supplementary Table 6

3) Corrected the interpretation accordingly.

4. In Supplemental Figure SF 1G, the authors did not find any changes in neutrophil numbers within the blood however they do not mention if and what changes they saw for neutrophils in the BAL samples. It would be helpful to show if and how the absolute numbers of BAL neutrophils correlate with viral titers and disease severity at the tested timepoints

6. Human studies show that patients with severe COVID-19 exhibit emergency myelopoiesis which is visible as presence of CD11b low/- neutrophils (Schulte-Schrepping et al. Cell 2020 and Silvin et al. Cell 2020). How did the surface CD11b levels look like for neutrophils (both BAL and blood) over the duration of infection? It would be great to show that the NHP models reflect similar changes as in humans, given that the current panel already has CD11b? If anything, it would add strength to the manuscript.

We could not analyze neutrophils in BAL because we did not include the CD66 marker in our panel. We have instead extended our analysis of the lungs, by staining new slides with MPO and CD11b Figure 4). We have included the two suggested outstanding papers in the references.

5. Line 151-152: This statement might need toning down since the authors present no direct evidence of these monocytes having anti-viral activity. It is implied but not proved.

We have the text modified accordingly (line 173).

7. Supplemental Figure SF 2I is also difficult to understand. A more detailed legend explaining how the PCA was plotted will help. We have included a detailed legend explaining which variables were used for the PCA and how it was calculated. In addition, we labeled each point in the PCA plot by animal ID (shape) and species (color) to aid interpretation.

8. Supplemental table 3 is difficult to understand. A more detailed legend will help. I am not sure what PC1-PC8 stands for? If these are principal components, how do they reflect what animal is what PC? Providing raw baseline values of each cytokine would be easier to comprehend. We included a detailed legend of the correlation coefficients shown in the table for each principal component and how to interpret the values. In addition, we added a supplementary table with raw values for each chemokine in order to increase clarity.

9. In Figure 1K: I am not sure how to read Figure 1K. There are 4 AGMs or RMs but a lot more dots. I assume each dot stands for chemokine levels in one animal? In that case maybe giving a different shape for each chemokine with species specific color will help? We updated the shapes and colors of each dot to specify animal ID and species as well as provided more details in the legend for clarity. We could not update the shapes to correspond with the chemokines because there were too many unique chemokines rendering the plot more difficult to interpret, so we added additional supplementary tables with both raw values and log₂ fold change data for better comprehension. (Supplementary table 7-9).

10. Line 167: "Overall this analysis revealed that animals with the most severe disease outcomes had higher levels of chemokines at week 1, including IP-10, TARC (CCL17) (Fig. 1L)". Does this stand true when AGM and RM analyze separately considering baseline differences for the NHPs?

The heat map represents the changes over baseline.

11. Figure 1L: Can the data for weeks 2,3,4 also be provided as in Fig 1L as supplemental data? It will help the readers decide the strength of the conclusions. **We have provided these data in Supplementary tables for all animals at all time points (see Supplementary table 7-9).**

12. In lines 185-199, the authors use the term macrophages to categorize the distinct myelomonocytic cells without use of markers that can discriminate between monocytes or macrophages. Monocytes also express CD11b and CD16 and considering that monocyte to macrophage transition is a continuum, it would be inaccurate to call these cells macrophages. I would suggest use of monocyte-derived cells or myeloid cells or myelomonocytic cells.

We have corrected the text accordingly.

13. Human studies show that patients with severe COVID-19 possess HLA-DR^{low}CD163^{hi} monocytes **in the blood**. How did surface CD163 levels change in macrophage subsets represented in Fig 2G-I at different timepoints? **While changes with respect to CD163 were too variable to be conclusive, we do see an increase in the frequency of CD14⁺ DR^{low} myeloid cells. We have now included this data in Supplementary Figure 2 and we also included the analysis for Arginase 1 activity, that we have shown being associated MDSCs and CD14⁺ HLADR cells in macaques.**

14. In Figure 3A. Moving the animal ID# off from the middle of the H&E images will make the image more visually appealing.

We have moved the animals ID and reformatted the Figure 3.

15. Figure 4: How do the frequency of NK cells, CD4 T cells, CD8 T cells, PD-1⁺ CD4 T cells, PD-1⁺ CD8 T cells correlate with viral burdens and disease severity. These analyses are now included in **Supplementary Figure 9.**

16. It will be useful to know where the distinct lymphocyte cells aggregate in the lungs of SARS-CoV2 infected NHPs. Do they intermingle with certain specific type of myeloid cells? Do they exist in certain parts of the lungs?

Aggregates are now shown in Supplementary Figure 6 and 7.

17. Figure 4G: Not sure if the tSNE accurately depicts the CD4 and CD8 regions. It makes comprehending the conclusion on Line 250 difficult. **We agree and added two more tSNE plots in the supplementary data to show the location of CD4⁺ and CD8⁺ T cells,**

18. Reference #14 doesn't really help the way it has been listed without title/journal/year. **We fixed it.**

19. Line 319: "For the first time, we identified TARC (CCL17) as central to myeloid recruitment in the lung during SARS-CoV-2." is inaccurate. There is no direct evidence (only correlation) for this in this work. This statement needs to be toned down as suggestion and not causative evidence.

We have modified the tone as suggested/line 402.

21. How does the age of the monkeys included in these studies compare with the age of humans? COVID-19 has worse outcomes in older human patients. A discussion on this would be helpful.

We have included information in material and methods. However, it is difficult to drive conclusions on the age of animals as compared to human age. Rhesus macaques live 20 years approximately in captivity and AGM live 16 years approximately. The age of AGM was approximated and GH99 was described by our vet as having a “old man lung”.

We have made all this information available to the reader at lines: 463-466.

Reviewers' Comments:

Reviewer #1:

Remarks to the Author:

The authors have comprehensively addresses the issues raised and substantially improved this important manuscript.

Reviewer #2:

Remarks to the Author:

The authors have addressed the concerns raised by the reviewers to the best of their ability.

Reviewer #3:

Remarks to the Author:

All my concerns have been addressed. No additional comments.

Point-by-point response to the reviewers

We would like to thank the reviewers for the great suggestions. We have addressed all the points as described below, and we have marked all the changes in the current manuscript in red.

Reviewer #1 (Remarks to the Author):

1. The abstract would benefit from a more substantial conclusion regarding the macrophages. What do these macrophage changes mean for pathology and disease? Why did they not include more of the overall results such as the CCL17?

We have now modified the abstract accordingly.

2. Macrophage gating is unclear. Was CD14 used to identify the macrophages? They demonstrate gating in figure 1 but should make the figures more clear as to what each population is and how it was defined (ie classical monocytes CD14++...)

We have included our gating strategy (Supplementary Figure 2A) and modified the nomenclature accordingly.

3. The IHC figures could all be better labeled so one can look at them and immediately understand what each stain/color is and doesn't have to reference the figure legends.

We have modified Figure 3 as suggested.

4. The IL-10 is very interesting and should be highlighted. Did the authors assess Tregs at all? They should at least discuss this as a possibility if it was not done originally.

We have now included the analysis for Arginase 1 (myeloid derived suppressor cells) and the tryptophan kynurenine ratio as a functional measure of T reg (IDO) activity (Figure 6D and Supplementary Fig 10) and expanded the discussion accordingly.

Reviewer #2 (Remarks to the Author):

Comments to authors: Here the authors have studied frequencies and phenotypes of myeloid cells in blood and lung of SARS-CoV-2 infected NHPs. The authors find elevated levels of proinflammatory monocytes which they argue is associated with disease severity. Overall the manuscript is interesting. However, there are several confounding variables that the authors need to control.

1. There were only 8 animals infected. The animals were of different species, different genders, different ages, and were infected via different routes. It is inappropriate to group them all together. They should be grouped according to their infection routes, species, age, and gender.

In our original submission we have been very careful to separate the AGM and RM by showing them in different colors and also by showing and analyzing them separately in the Supplementary Figures. We have now provided additional statistical analysis showing that no differences in disease severity nor viral replication could be found between females and males nor challenge routes (Supplementary Table 4). We think the range of the animal age (13-16) is not large enough that we can stratify accordingly.

Even with these, much of the statistical variation is attributed to the outlier animal (GH99). We have now provided evidence that GH99 is not driving the main conclusions of our analysis by including additional figures and statistics in Supplementary Figures 1G, 1M, and 3K where this animal was removed. Please note that all the analysis retained significance when this animal is excluded.

2. The authors should show longitudinal pulse oxygen levels and temperatures for all of the animals. We thank the reviewer for the suggestion. Information can now be found in Supplementary Table 3 (lines 130 -134).

3. The authors should provide representative flow gating. It's possible there are contaminating NK cells in their 14 vs 16 plots. Moreover, myeloid cells from the lung (BAL in particular) are notoriously difficult to study by flow cytometry. We have included gating strategy as suggested by the reviewer (Supplementary Figure 2A and Figure 7A). Myelomonocytic cells gated in BAL were originally shown in Supplementary Figure 3 which is now SF4.

4. are different numbers available for week 1 timepoints for the phenotypic comparisons, but not for the volumetric counts (Figure 1)?

We have now provided absolute counts for the subsets in Supplementary Figure 2. We have provided numbers of animals in figure legends.

5. Why are data from NC34 available from week 2 for VL (1B), but no other data?

NC34 was euthanized on week 1, day 8 because of respiratory distress as stated at line 117-118.

6. The utility of the IL10/IL6 ratio is unclear given many of the animals had undetected IL10 (5/8 at necropsy to this reviewer's eyes). There's clearly a problem with the AGM/RM symbols in figure 5B/C also. (in B three AGM have IL10/IL6 ratios of 0, but in C one AGM and 2 rhesus have IL10/IL6 ratios of 0).

We agree more studies are needed to understand the utility of the IL-10 /IL-6 ratio as we have stated in the discussion, we have corrected the symbols and per request by the other two reviewers we now have elaborated on this point by adding ulterior information regarding the possible role of T regulatory cells /myeloid derived suppressor cells (reviewer1) and added analysis of Tryptophan catabolism as a measurement for IDO/treg activity (Figure 6D supplementary Figure 10).

Reviewer #3 (Remarks to the Author):

In the given manuscript, Fahlberg et al. present convincing evidence regarding the dynamics of distinct immune cells within the lungs and blood of 2 non-human primate (NHPs) models namely African green monkeys (AGM) and Rhesus macaques (RM) at distinct time points of the SARS-CoV2 infection. This is a well-controlled, statistically-sound, fascinating and significant study and is a very well written manuscript which recapitulates a lot of findings recently reported for human COVID-19 patients in NHPs.

The manuscripts are Schulte-Schrepping et al. Cell 2020 (doi:<https://doi.org/10.1016/j.cell.2020.08.001>) and Silvin et al. Cell 2020 (doi:<https://doi.org/10.1016/j.cell.2020.08.002>).

Having said that the three pieces of novelty that this manuscript brings to the community is (i) the inclusion of exhaustive time points in the study, (ii) the identification of CCL17 as a potential mediator of myeloid recruitment to the lungs during COVID-19 and (iii) potential use of IL-10 to IL-6 ratios in predicting outcomes and course of COVID-19 disease; all these studies being done in AGMs and RMs that have previously been shown to be good models of SARS-CoV2 infection by the authors and others.

Given the recent precedence for their observations in human studies, there are several questions that should be addressed for a more comprehensive presentation of this study. Also considering the expenses associated with doing new NHP studies and the COVID-19 related significance of this work, suggested concerns have been carefully chosen to be address-able with the existing datasets and samples.

The comments in order of their mention in the manuscript are:

1. Line 39: "...and validate NHP as models to test immune therapies" is inaccurate. The results do not validate NHPs as models to test immune therapies but suggest that they could make good models for the same. Validation of therapies will need a lot larger sample size and use of candidate therapies both of which were not within the scope of this study. **We agree with this reviewer and we have modified the text accordingly in Abstract and Introduction (last paragraph).**

2. In Figure 1, it would be interesting to know if and how the absolute numbers of classical and non-classical monocytes correlate with viral titers at the tested timepoints.

3. Along the same lines, in Figure 1, How do the absolute numbers of classical and non-classical monocytes correlate with the disease severity?

In Figure 1H and 1I, HD09 and GH99 have the least non-classical monocytes and most classical monocyte numbers and these also happen to be the RMs with the worst disease severity (Table 1). This could suggest that non-classical monocytes are reflective of worse outcomes which is in lines with the published human data.

This point 2 also makes it inaccurate for the conclusion made **on Line 150**: "Pre-infection levels were restored at 3 weeks post-infection (week 1 vs. week 3: classical $P = 0.03$ and non-classical $P = 0.03$). Clearly HD09 and GH99 (the RMs with worse outcomes) never regained their baseline numbers of non-classical monocytes.

These reasons make it necessary to study the cellular effects that the authors have reported in with stratification of NHPs with worse vs milder outcomes, albeit in supplemental data with a smaller sample size. It would be a phenomenal addition to the already existing dataset.

We thank the reviewer for these suggestions.

We have now:

1) included the absolute numbers for the monocytic subsets in Supplementary Figure 2H-I (and line 166-183)

2) Described associations with frequency and numbers and viral load /disease and add a new Supplementary Table 6

3) Corrected the interpretation accordingly.

4. In Supplemental Figure SF 1G, the authors did not find any changes in neutrophil numbers within the blood however they do not mention if and what changes they saw for neutrophils in the BAL samples. It would be helpful to show if and how the absolute numbers of BAL neutrophils correlate with viral titers and disease severity at the tested timepoints

6. Human studies show that patients with severe COVID-19 exhibit emergency myelopoiesis which is visible as presence of CD11b low/- neutrophils (Schulte-Schrepping et al. Cell 2020 and Silvin et al. Cell 2020). How did the surface CD11b levels look like for neutrophils (both BAL and blood) over the duration of infection? It would be great to show that the NHP models reflect similar changes as in humans, given that the current panel already has CD11b? If anything, it would add strength to the manuscript.

We could not analyze neutrophils in BAL because we did not include the CD66 marker in our panel. We have instead extended our analysis of the lungs, by staining new slides with MPO and CD11b Figure 4). We have included the two suggested outstanding papers in the references.

5. Line 151-152: This statement might need toning down since the authors present no direct evidence of these monocytes having anti-viral activity. It is implied but not proved.

We have the text modified accordingly (line 173).

7. Supplemental Figure SF 2I is also difficult to understand. A more detailed legend explaining how the PCA was plotted will help. We have included a detailed legend explaining which variables were used for the PCA and how it was calculated. In addition, we labeled each point in the PCA plot by animal ID (shape) and species (color) to aid interpretation.

8. Supplemental table 3 is difficult to understand. A more detailed legend will help. I am not sure what PC1-PC8 stands for? If these are principal components, how do they reflect what animal is what PC? Providing raw baseline values of each cytokine would be easier to comprehend. We included a detailed legend of the correlation coefficients shown in the table for each principal component and how to interpret the values. In addition, we added a supplementary table with raw values for each chemokine in order to increase clarity.

9. In Figure 1K: I am not sure how to read Figure 1K. There are 4 AGMs or RMs but a lot more dots. I assume each dot stands for chemokine levels in one animal? In that case maybe giving a different shape for each chemokine with species specific color will help? We updated the shapes and colors of each dot to specify animal ID and species as well as provided more details in the legend for clarity. We could not update the shapes to correspond with the chemokines because there were too many unique chemokines rendering the plot more difficult to interpret, so we added additional supplementary tables with both raw values and log2 fold change data for better comprehension. (Supplementary table 7-9).

10. Line 167: "Overall this analysis revealed that animals with the most severe disease outcomes had higher levels of chemokines at week 1, including IP-10, TARC (CCL17) (Fig. 1L)". Does this stand true when AGM and RM analyze separately considering baseline differences for the NHPs?

The heat map represents the changes over baseline.

11. Figure 1L: Can the data for weeks 2,3,4 also be provided as in Fig 1L as supplemental data? It will help the readers decide the strength of the conclusions. **We have provided these data in Supplementary tables for all animals at all time points (see Supplementary table 7-9).**

12. In lines 185-199, the authors use the term macrophages to categorize the distinct myelomonocytic cells without use of markers that can discriminate between monocytes or macrophages. Monocytes also express CD11b and CD16 and considering that monocyte to macrophage transition is a continuum, it would be inaccurate to call these cells macrophages. I would suggest use of monocyte-derived cells or myeloid cells or myelomonocytic cells.

We have corrected the text accordingly.

13. Human studies show that patients with severe COVID-19 possess HLA-DR^{low}CD163^{hi} monocytes **in the blood**. How did surface CD163 levels change in macrophage subsets represented in Fig 2G-I at different timepoints? **While changes with respect to CD163 were too variable to be conclusive, we do see an increase in the frequency of CD14⁺ DR^{low} myeloid cells. We have now included this data in Supplementary Figure 2 and we also included the analysis for Arginase 1 activity, that we have shown being associated MDSCs and CD14⁺ HLADR cells in macaques.**

14. In Figure 3A. Moving the animal ID# off from the middle of the H&E images will make the image more visually appealing.

We have moved the animals ID and reformatted the Figure 3.

15. Figure 4: How do the frequency of NK cells, CD4 T cells, CD8 T cells, PD-1⁺ CD4 T cells, PD-1⁺ CD8 T cells correlate with viral burdens and disease severity. These analyses are now included in **Supplementary Figure 9.**

16. It will be useful to know where the distinct lymphocyte cells aggregate in the lungs of SARS-CoV2 infected NHPs. Do they intermingle with certain specific type of myeloid cells? Do they exist in certain parts of the lungs?

Aggregates are now shown in Supplementary Figure 6 and 7.

17. Figure 4G: Not sure if the tSNE accurately depicts the CD4 and CD8 regions. It makes comprehending the conclusion on Line 250 difficult. **We agree and added two more tSNE plots in the supplementary data to show the location of CD4⁺ and CD8⁺ T cells,**

18. Reference #14 doesn't really help the way it has been listed without title/journal/year. **We fixed it.**

19. Line 319: "For the first time, we identified TARC (CCL17) as central to myeloid recruitment in the lung during SARS-CoV-2." is inaccurate. There is no direct evidence (only correlation) for this in this work. This statement needs to be toned down as suggestion and not causative evidence.

We have modified the tone as suggested/line 402.

21. How does the age of the monkeys included in these studies compare with the age of humans? COVID-19 has worse outcomes in older human patients. A discussion on this would be helpful.

We have included information in material and methods. However, it is difficult to drive conclusions on the age of animals as compared to human age. Rhesus macaques live 20 years approximately in captivity and AGM live 16 years approximately. The age of AGM was approximated and GH99 was described by our vet as having a “old man lung”.

We have made all this information available to the reader at lines: 463-466.